# Inverse Q-Learning Done Right:
# Offline Imitation Learning in $Q^\pi$-Realizable MDPs

**Antoine Moulin**
Universitat Pompeu Fabra
antoine.moulin@upf.edu

**Gergely Neu**
Universitat Pompeu Fabra
gergely.neu@gmail.com

**Luca Viano**
EPFL
luca.viano@epfl.ch

## Abstract

We study the problem of offline imitation learning in Markov decision processes (MDPs), where the goal is to learn a well-performing policy given a dataset of state-action pairs generated by an expert policy. Complementing a recent line of work on this topic that assumes the expert belongs to a tractable class of known policies, we approach this problem from a new angle and leverage a different type of structural assumption about the environment. Specifically, for the class of linear $Q^\pi$-realizable MDPs, we introduce a new algorithm called saddle-point offline imitation learning (SPOIL), which is guaranteed to match the performance of any expert up to an additive error $\varepsilon$ with access to $\mathcal{O}(\varepsilon^{-2})$ samples. Moreover, we extend this result to possibly nonlinear $Q^\pi$-realizable MDPs at the cost of a worse sample complexity of order $\mathcal{O}(\varepsilon^{-4})$. Finally, our analysis suggests a new loss function for training critic networks from expert data in deep imitation learning. Empirical evaluations on standard benchmarks demonstrate that the neural net implementation of SPOIL is superior to behavior cloning and competitive with state-of-the-art algorithms.

## 1 Introduction

In imitation learning (IL), a learner observes a finite dataset of state-action pairs generated by an expert policy interacting with an environment modeled as a Markov Decision Process (MDP; Puterman, 2014). The learner's objective is to find a policy that performs nearly as well as the expert policy with respect to an unknown ground-truth reward function. This work focuses on *offline imitation learning*, where the learner cannot collect new state-action sequences from the MDP used for generating the expert's data. In this context, we propose new algorithms that operate under a previously under-explored set of structural assumptions on the learning environment.

Recent years have seen a quite significant surge of interest in the problem of imitation learning, not unlikely due to its relevance to next-token prediction in generative language models (Rajaraman et al., 2020; Foster et al., 2024; Rohatgi et al., 2025). A common feature of these recent works is that they all make the assumption that the expert data has been generated by a fixed policy that belongs to a known, finite class of policies and they return policies within the same class. Such an assumption is often referred to as *expert realizability* and can be formally stated as follows.

**Assumption** (Expert realizability). *The learner has access to a function class $\Pi^{\text{E}}$ that contains the unknown expert policy $\pi_{\text{E}}$, that is, such that $\pi_{\text{E}} \in \Pi^{\text{E}}$.*

Several clean and elegant results were proved under this assumption, in particular showing the existence of conceptually simple algorithms achieving tight upper bounds on the sample complexity of finding good solutions, and lower bounds demonstrating the near-optimality of these algorithms under said assumptions. These bounds typically depend on a measure of complexity of the policy class (as measured by, say, its covering number). However, further scrutiny reveals that these assumptions

39th Conference on Neural Information Processing Systems (NeurIPS 2025).

may not always be verified or even reasonable: in many cases of significant practical interest, there is no reason to believe that the expert policy may be easily modeled within a simple and tractable policy class. For instance, in the popular use case of learning from human feedback, it is arguably quite unlikely that data would be generated in a consistent, systematically predictable way that can be modeled as a simple policy mapping states to actions. Indeed, human behavior can be nonstationary, irrational, or even be influenced by unobserved confounders not captured by the state representation. We address these limitations by exploring an alternative framework for imitation learning, which reasons about the structure of the *value functions* of the policies used by the *learning algorithm* itself, as opposed to making assumptions about the structure of the policy followed by the expert.

Furthermore, the sample complexity guarantees in Rajaraman et al. (2020); Foster et al. (2024); Rohatgi et al. (2025) scale with $\log |\Pi^{\mathrm{E}}|$ (assuming $\Pi^{\mathrm{E}}$ is finite), meaning large policy classes, potentially necessary to realize the expert, lead to deteriorated guarantees. Additionally, the consequences of misspecification, *i.e.*, $\pi_{\mathrm{E}} \notin \Pi^{\mathrm{E}}$, are often severe. For instance, Rohatgi et al. (2025) demonstrated that if the policy class $\Pi^{\mathrm{E}}$ is misspecified, then it is computationally intractable to learn $\arg\min_{\pi \in \Pi^{\mathrm{E}}} \mathcal{D}_{\mathsf{H}}^2(\mathbb{P}^{\pi}, \mathbb{P}^{\pi_{\mathrm{E}}})$, the best in-class policy under the Hellinger distance, in an offline manner. However, this theoretical intractability under misspecification seems at odds with practical scenarios, such as training large language models via next-token prediction (a form of offline IL), which perform well despite the expert policy (derived from human-written text) likely not belonging to any reasonable policy class $\Pi^{\mathrm{E}}$.

To address this apparent discrepancy, we initiate the study of offline IL by leveraging structural assumptions about the MDP rather than relying on expert realizability. For example, in language tasks, structural assumptions might involve deterministic, tree-shaped MDPs. In robotics, one might assume that next states are determined by compact feature representations of current state-action pairs. More generally, we consider MDPs where the action-value functions of a subset of policies can be written as a linear combination of features known to the learner. Such MDPs are referred to as linear $Q^{\pi}$-realizable MDPs, a class that has been central to recent works in reinforcement learning theory (Weisz et al., 2023; Mhammedi, 2025; Tkachuk et al., 2024). Our primary contribution is to show that, for this class of MDPs, it is possible to develop algorithms that guarantee to output a policy performing arbitrarily close to the expert policy *without imposing expert realizability*.

The algorithm is based on a simple primal-dual formulation of the problem of imitation learning, which characterizes the solution as the saddle-point of a convex-concave objective function. The primal variables correspond to policies in the MDP and the dual variables to Q-functions, which motivates a very simple saddle-point optimization algorithm for imitation learning: in a sequence of rounds, the primal player (the *actor*) picks a policy and the dual player (the *critic*) picks a Q-function, respectively trying to minimize and maximize the objective. We accordingly call the method SPOIL, standing for Saddle-Point Offline Imitation Learning. In the case of linear function approximation, both update steps of SPOIL can be performed very efficiently (in time linear in the feature dimension). For general function approximation, the Q-function updates can be performed by solving a simple linear optimization problem, which is straightforward to solve in practical scenarios. When instantiated with neural networks, empirical experiments show its performance is competitive with (and in some cases superior to, *e.g.*, behavior cloning) state-of-the-art offline imitation learning algorithms. Interestingly, our algorithm shares a good degree of similarity with the state-of-the-art method of Garg et al. (2021) called IQ-Learn, which is also derived from a primal-dual perspective. We discuss these similarities in depth and argue that SPOIL provides a superior solution to the IQ-Learn objective (at least inasmuch as it is more amenable to theoretical analysis).

To the best of our knowledge, this is the first result showing that leveraging structural assumptions of the underlying MDP can guarantee matching the expert performance as the number of expert transitions goes to infinity without imposing any form of expert realizability assumption. For clarity, we compare our contribution with existing results in Table 1. We denoted $\mathcal{N}_{\epsilon}(\mathcal{Q}, \|\cdot\|_{\infty})$ the $\epsilon$-covering number of the function class $\mathcal{Q}$ (see Theorem 2), and $\tau_{\mathrm{E}}$ the number of trajectories needed to make the difference in total expected return between the expert and the output policy smaller than $\varepsilon$.

**Notation.** We use $\Delta(\mathcal{Z})$ to denote the simplex over the countable set $\mathcal{Z}$. Given two probability distributions $p, q \in \Delta(\mathcal{Z})$, we denote the Kullback-Leibler divergence as $\mathcal{D}_{\mathsf{KL}}(p, q) = \sum_{z \in \mathcal{Z}} p(z) \log \frac{p(z)}{q(z)}$. We denote $\langle \cdot, \cdot \rangle$ the inner product between two finite-dimensional vectors, and $\|\cdot\|$ the Euclidean norm. We denote $\mathcal{U}([K])$ the uniform distribution over the set $[K] = \{1, \ldots, K\}$. The Euclidean ball of radius $R > 0$ centered at the origin is denoted as $\mathfrak{B}(R)$.

Table 1: Comparison with related algorithms. We denoted the class of deterministic linear experts as $\Pi_{\text{det, lin}}^{\text{E}} = \left\{ \pi : \exists \theta \in \mathfrak{B}(B_\theta), \pi(\cdot) = \arg\max_{a \in \mathcal{A}} \langle \theta, \varphi(\cdot, a) \rangle \right\}$, and an arbitrary policy class as $\Pi^{\text{E}}$. We also define $W = \max_{\pi \in \Pi^{\text{E}}, h \in [H]} \left\| \frac{\pi_{\text{E},h}}{\pi_h} \right\|_\infty$, $\varepsilon_{\text{miss}} = \min_{\pi \in \Pi^{\text{E}}} \mathcal{D}_{\text{H}}^2(\mathbb{P}^\pi, \mathbb{P}^{\pi_{\text{E}}})$, and $\varepsilon' = \tilde{\mathcal{O}}(\varepsilon^3)$.

| Algorithm | Structural assumptions | Avoids expert realizability | Infinite horizon | Expert class | Expert Traj. $(\tau_{\text{E}})$ |
|---|---|---|---|---|---|
| BC with log loss (Foster et al., 2024) | – | ✗ | ✗ | $\Pi^{\text{E}}$ | $\mathcal{O}\left( \frac{H^2 \log|\Pi^{\text{E}}|}{\varepsilon^2} \right)$ |
| BC with 0-1 loss (Rajaraman et al., 2021) | – | ✗ | ✗ | $\Pi_{\text{det, lin}}^{\text{E}}$ | $\tilde{\mathcal{O}}\left( \frac{H^2 d}{\varepsilon} \right)$ |
| BoostedLogLossBC (Rohatgi et al., 2025) | – | ✓ with a misspecification error of $\tilde{\mathcal{O}}(H \log(W) \varepsilon_{\text{miss}})$ | ✗ | $\Pi^{\text{E}}$ | $\mathcal{O}\left( \frac{H^2 \log|\Pi^{\text{E}}|}{\varepsilon^2} \right)$ |
| Projection (Abbeel and Ng, 2004) | Linear reward Known transitions | ✓ | ✓ | – | $\tilde{\mathcal{O}}\left( \frac{d}{(1-\gamma)^2 \varepsilon^2} \right)$ |
| MWAL (Syed and Schapire, 2007) | Linear reward Known transitions | ✓ | ✓ | – | $\tilde{\mathcal{O}}\left( \frac{\log(d)}{(1-\gamma)^2 \varepsilon^2} \right)$ |
| SPOIL (Theorem 1) | Linear $Q^\pi$-realizability | ✓ | ✓ | – | $\tilde{\mathcal{O}}\left( \frac{d}{(1-\gamma)^4 \varepsilon^2} \right)$ |
| SPOIL (Theorem 2) | $Q^\pi$-realizability | ✓ | ✓ | – | $\tilde{\mathcal{O}}\left( \frac{\log \mathcal{N}_{\varepsilon'}(\mathcal{Q}, \|\cdot\|_\infty)}{(1-\gamma)^8 \varepsilon^4} \right)$ |

## 2 Preliminaries

We begin by introducing the problem of offline imitation learning in discounted MDPs together with the assumptions we will consider throughout the paper.

**Markov decision processes.** We formalize the learning problem in a discounted MDP $\mathcal{M} = (\mathcal{X}, \mathcal{A}, r, P, \gamma, \nu_0)$, where $\mathcal{X}$ is the state space which we assume finite but too large to be enumerated, $\mathcal{A}$ is a finite action space with $A$ actions, $r: \mathcal{X} \times \mathcal{A} \to [0, 1]$ is the unknown reward function, $P: \mathcal{X} \times \mathcal{A} \to \Delta(\mathcal{X})$ is the unknown transition kernel, $\gamma \in [0, 1)$ is the discount factor, and $\nu_0 \in \Delta(\mathcal{X})$ is the initial state distribution. For any state-action-state triplet $(x, a, x')$, $P(x' \mid x, a)$ denotes the probability of landing in state $x'$ after taking action $a$ in state $x$. A *stationary policy* (or simply *policy*) $\pi: \mathcal{X} \to \Delta(\mathcal{A})$ is a mapping from states to distributions over actions. The interaction of a policy $\pi$ with the environment $\mathcal{M}$ unfolds as follows: an initial state $X_0 \sim \nu_0$ is drawn, and for each subsequent time step $h \geq 0$, an action $A_h \sim \pi(\cdot \mid X_h)$ is taken, a reward $r(X_h, A_h)$ is received, and the agent transitions to a new state $X_{h+1} \sim P(\cdot \mid X_h, A_h)$. We denote $\mathbb{P}^\pi$ the resulting probability distribution over trajectories, and $\mathbb{E}^\pi$ the corresponding expectation operator. For any state $x \in \mathcal{X}$, we define the state value function of the policy $\pi$ as $V^\pi(x) = \mathbb{E}^\pi\left[ \sum_{h=0}^\infty \gamma^h r(X_h, A_h) \mid X_0 = x \right]$. Analogously, we define the state-action value function as $Q^\pi(x, a) = \mathbb{E}^\pi\left[ \sum_{h=0}^\infty \gamma^h r(X_h, A_h) \mid X_0 = x, A_0 = a \right]$. The value functions are tied together via the *Bellman equations*

$$V^\pi(x) = \sum_{a \in \mathcal{A}} \pi(a \mid x) Q^\pi(x, a), \quad \text{and} \quad Q^\pi(x, a) = r(x, a) + \gamma \sum_{x' \in \mathcal{X}} P(x' \mid x, a) V^\pi(x').$$

Additionally, we will sometimes use the notation $Q(x, \pi)$ to denote $\sum_a \pi(a \mid x) Q(x, a)$ for any policy $\pi$ and any function $Q: \mathcal{X} \times \mathcal{A} \to \mathbb{R}$. Note that this notation allows us to write $V^\pi(x) = Q^\pi(x, \pi)$. Any policy $\pi$ induces an *occupancy measure* $\mu^\pi \in \Delta(\mathcal{X} \times \mathcal{A})$ over state-action pairs, defined as the discounted total expected times that each state-action pair is visited by policy $\pi$. The same quantity defined for states is called the state-occupancy measure and is denoted as $\nu^\pi \in \Delta(\mathcal{X})$. For any state-action pair $(x, a) \in \mathcal{X} \times \mathcal{A}$, they are respectively defined as

$$\nu^\pi(x) = (1 - \gamma) \sum_{h=0}^\infty \gamma^h \mathbb{P}^\pi[X_h = x], \quad \text{and} \quad \mu^\pi(x, a) = (1 - \gamma) \sum_{h=0}^\infty \gamma^h \mathbb{P}^\pi[X_h = x, A_h = a],$$

and they are related to each other by the *flow conditions* (sometimes called "Bellman flow conditions")

$$\nu^\pi(x) = \gamma \sum_{x', a'} P(x \mid x', a') \mu^\pi(x', a') + (1 - \gamma) \nu_0(x). \tag{1}$$

Notably, these definitions and the flow conditions remain valid for general history-dependent policies $\pi$ that may take the entire history of state-action pairs $(X_1, A_1, \ldots, X_h)$ into account when selecting each action $A_h$. Finally, we let $\rho^{\pi} = (1 - \gamma)\mathbb{E}^{\pi}\left[\sum_{h=0}^{\infty} \gamma^h r(X_h, A_h)\right]$ stand for the normalized expected return of a (potentially nonstationary) policy $\pi$. The following useful result, commonly called the *performance-difference lemma* (Kakade and Langford 2002, see also Eq. 7.14 in Howard 1960), gives a useful expression for the performance gap between two policies.

**Lemma 1.** *Let $\pi$ be a stationary policy and $\pi'$ be any policy. Then,*

$$\rho^{\pi'} - \rho^{\pi} = \mathbb{E}_{(X,A)\sim\mu^{\pi'}}[Q^{\pi}(X, A) - V^{\pi}(X)].$$

Note that this lemma is generally stated for stationary policies, but we will find it useful later to use it with general history-dependent policies. We provide the straightforward proof in Appendix B.1.

**Imitation Learning.** We consider the problem of offline imitation learning. Given a dataset $\mathcal{D}^{\pi_{\mathrm{E}}} = \left\{X_{\mathrm{E}}^i, A_{\mathrm{E}}^i\right\}_{i=1}^{\tau_{\mathrm{E}}}$ of state-action pairs sampled from an expert policy's occupancy measure $\mu^{\pi_{\mathrm{E}}}$, our objective is to design an algorithm, Alg, that produces a policy $\pi^{\mathrm{out}}$ satisfying

$$\mathbb{E}\left[\rho^{\pi_{\mathrm{E}}} - \rho^{\pi^{\mathrm{out}}}\right] \leq \varepsilon. \tag{2}$$

The algorithm is not allowed any further interaction with the expert policy or the MDP $\mathcal{M}$ and only has to work with the record of state-action pairs contained in the data set. As stated in the introduction, we aim to achieve this *without imposing expert realizability*. Instead, we consider the following structural assumption on the environment.

**Assumption 1** (Linear $Q^{\pi}$-realizability). *Let $B_{\theta}, B_{\varphi} > 0$. Given a known mapping $\varphi\colon \mathcal{X} \times \mathcal{A} \to \mathbb{R}^d$, consider the policy class $\Pi_{\mathrm{lin}}$ defined as follows*

$$\Pi_{\mathrm{lin}} = \left\{\pi \in \Delta(\mathcal{A})^{\mathcal{X}} : \exists(\theta_k)_{k\in[K]} \subset \mathfrak{B}(B_{\theta}), \pi(a \,|\, x) = \frac{\exp\left(\eta \sum_{k=1}^K \langle \varphi(x,a), \theta_k \rangle\right)}{\sum_{b\in\mathcal{A}} \exp\left(\eta \sum_{k=1}^K \langle \varphi(x,b), \theta_k \rangle\right)}\right\}.$$

*For any policy $\pi \in \Pi_{\mathrm{lin}}$, there exists a vector $\theta^{\pi} \in \mathfrak{B}(B_{\theta})$ such that for any state-action pair $(x, a)$, $Q^{\pi}(x, a) = \langle \varphi(x,a), \theta^{\pi} \rangle$. Besides, assume $\sup_{x,a} \|\varphi(x,a)\| \leq B_{\varphi}$, and $\sup_{x,a} \sup_{\theta\in\mathfrak{B}(B_{\theta})} \langle \varphi(x,a), \theta \rangle \leq \frac{1}{1-\gamma}$.*

Notice that we need to assume only linearity of the state action value function for the class of softmax linear policies $\Pi_{\mathrm{lin}}$. In contrast, prior works on linear $Q^{\pi}$-realizable MDPs (Weisz et al., 2023; Mhammedi, 2025) require the above assumption to hold for all Markov policies. Moreover, we highlight that we potentially have that $\pi_{\mathrm{E}} \notin \Pi_{\mathrm{lin}}$, therefore we do not require realizability of the expert state action value function.

We will also consider the general function approximation setting, where the action value function of any policy $\pi$ can be represented by some function class $\mathcal{Q} \subset \mathbb{R}^{\mathcal{X}\times\mathcal{A}}$.

**Assumption 2** ($Q^{\pi}$-realizability). *An MDP is said $Q^{\pi}$-realizable if there exists a function class $\mathcal{Q} \subset \mathbb{R}^{\mathcal{X}\times\mathcal{A}}$ such that for any policy $\pi \in \Pi_{\mathcal{Q}}$ defined as*

$$\Pi_{\mathcal{Q}} = \left\{\pi \in \Delta(\mathcal{A})^{\mathcal{X}} : \exists(Q_k)_{k\in[K]} \subset \mathcal{Q}, \pi(a \,|\, x) = \frac{\exp\left(\eta \sum_{k=1}^K Q_k(x,a)\right)}{\sum_{b\in\mathcal{A}} \exp\left(\eta \sum_{k=1}^K Q_k(x,b)\right)}\right\},$$

*it holds that $Q^{\pi} \in \mathcal{Q}$, and for any $Q \in \mathcal{Q}$, $\|Q\|_{\infty} \leq \frac{1}{1-\gamma}$.*

For this assumption to make sense, we typically require the function class $\mathcal{Q}$ to have bounded capacity. We quantify this via covering numbers, defined as follows.

**Definition 1** (Covering number). *Let $(M, d)$ be a metric space, $K$ be a subset of $M$, and $\epsilon > 0$. A set $\mathcal{C}_{\epsilon}(K, d)$ is an $\epsilon$-covering of $K$ if for any $x \in K$, there exists $y \in \mathcal{C}_{\epsilon}(K, d)$ such that $d(x, y) \leq \epsilon$. The covering number of $K$, $\mathcal{N}_{\epsilon}(K, d)$, is the minimum cardinality of any such covering of $K$.*

## 3 Primal-dual offline imitation learning

In order to introduce our main algorithmic idea, we define the following objective function:

$$\mathcal{L}(\pi; Q) = \mathbb{E}_{(X,A) \sim \mu^{\pi_{\text{E}}}} [Q(X, A) - Q(X, \pi)],$$

where we denoted $Q(X, \pi) = \mathbb{E}_{A' \sim \pi(\cdot \mid X)}[Q(X, A')]$. Our main observation is that the main objective function we consider can be rewritten in terms of this function as follows:

$$\rho^{\pi_{\text{E}}} - \rho^{\pi} = \mathcal{L}(\pi; Q^{\pi}) \leq \sup_{Q \in \mathcal{Q}} \mathcal{L}(\pi; Q).$$

This suggests a good policy $\pi^{\text{out}}$ may be found by solving the *saddle-point* optimization problem $\min_{\pi} \sup_{Q \in \mathcal{Q}} \mathcal{L}(\pi; Q)$. Indeed, if one is able to produce a policy $\pi^{\text{out}}$ satisfying $\sup_{Q \in \mathcal{Q}} \mathcal{L}(\pi^{\text{out}}; Q) \leq \varepsilon$, then the above inequality implies that the suboptimality of $\pi^{\text{out}}$ as compared to $\pi_{\text{E}}$ will also be at most $\varepsilon$.

Inspired by this observation, we set out to design an incremental *primal-dual* optimization algorithm to approximate the saddle point of the function $\mathcal{L}$. In each iteration $k = 1, 2, \ldots, K$, the algorithm performs two updates: a primal update that corresponds to policy updates aiming to minimize $\mathcal{L}$, and a dual update that computes action-value function estimates and aims to maximize $\mathcal{L}$. Following a common terminology in reinforcement learning, we will sometimes refer to the primal updates as *actor* updates and the dual updates as *critic* updates.

In order to turn these insights into a practical algorithm, we define the following empirical estimate of the objective function $\mathcal{L}$:

$$\widehat{\mathcal{L}}(\pi; Q) = \frac{1}{\tau_{\text{E}}} \sum_{i=1}^{\tau_{\text{E}}} \left( Q(X_{\text{E}}^i, A_{\text{E}}^i) - Q(X_{\text{E}}^i, \pi) \right).$$

For a fixed $Q$ and $\pi$, this is clearly an unbiased estimator of $\mathcal{L}$. In line with the derivations above, we choose our critic and actor updates respectively as

$$Q_k \in \underset{Q \in \mathcal{Q}}{\arg\max}\, \widehat{\mathcal{L}}(\pi_k; Q), \quad \text{and} \quad \pi_{k+1}(a \mid x) = \frac{\pi_k(a \mid x) e^{\eta Q_k(x,a)}}{\sum_{a' \in \mathcal{A}} \pi_k(a' \mid x) e^{\eta Q_k(x,a')}},$$

where $\eta > 0$ is a *learning-rate* (or *stepsize*) parameter that modulates the strength of the policy updates. After performing $K$ updates, the algorithm chooses a random index $I$ uniformly on the integers in $[\![1, K]\!]$, and returns $\pi^{\text{out}} = \pi_I$. We refer to this algorithm as Saddle-Point Offline Imitation Learning (SPOIL). This algorithm design is justified by the following simple error decomposition that lies at the heart of our main results.

**Proposition 1.** *Let $\Delta(\pi) = \sup_{Q \in \mathcal{Q}} \left| \mathcal{L}(\pi; Q) - \widehat{\mathcal{L}}(\pi; Q) \right|$. The output of SPOIL satisfies*

$$\mathbb{E}\left[ \rho^{\pi_{\text{E}}} - \rho^{\pi^{\text{out}}} \right] \leq \frac{1}{K} \sum_{k=1}^{K} \mathbb{E}[\mathcal{L}(\pi_k; Q_k)] + \frac{2}{K} \sum_{k=1}^{K} \mathbb{E}[\Delta(\pi_k)].$$

*Proof.* The proof simply follows by noticing

$$\mathbb{E}\left[ \rho^{\pi_{\text{E}}} - \rho^{\pi^{\text{out}}} \right] = \frac{1}{K} \sum_{k=1}^{K} \mathbb{E}[\mathcal{L}(\pi_k; Q^{\pi_k})] \leq \frac{1}{K} \sum_{k=1}^{K} \mathbb{E}\left[ \widehat{\mathcal{L}}(\pi_k; Q^{\pi_k}) \right] + \frac{1}{K} \sum_{k=1}^{K} \mathbb{E}[\Delta(\pi_k)]$$

$$\leq \frac{1}{K} \sum_{k=1}^{K} \mathbb{E}\left[ \widehat{\mathcal{L}}(\pi_k; Q_k) \right] + \frac{1}{K} \sum_{k=1}^{K} \mathbb{E}[\Delta(\pi_k)] \leq \frac{1}{K} \sum_{k=1}^{K} \mathbb{E}[\mathcal{L}(\pi_k; Q_k)] + \frac{2}{K} \sum_{k=1}^{K} \mathbb{E}[\Delta(\pi_k)],$$

where we have used the definitions of $\Delta$ and $Q_k$ in the first and second lines, respectively. $\square$

The first term in this decomposition corresponds to the *regret* of the policy player $\pi$ against the comparator strategy $\pi_{\text{E}}$ and can be controlled with probability 1 via standard tools of online learning (as found in the excellent books of Cesa-Bianchi and Lugosi 2006 and Orabona 2023). The second term measures the estimation error of the objective function $\mathcal{L}$ uniformly over the space of action-value functions $\mathcal{Q}$ and along the policies played by the algorithm, and can be controlled with high

probability via standard concentration arguments. Altogether, the proposition suggests that `SPOIL` will return a good policy if these estimation errors can be bounded reasonably—a fact we will formally show in the next section.

Before stating our performance guarantees for the concrete settings we consider in this paper, we pause to point out a peculiar connection between the algorithm described above and the inverse Q-learning (`IQ-Learn`) algorithm of Garg et al. (2021). While motivated using completely different arguments, the saddle-point objective function optimized by `IQ-Learn` is nearly identical to our function $\mathcal{L}$: after removing entropy-regularization and setting their reward regularizer $\psi$ to zero, one can verify using the flow constraint (Eq. 1) that their function $\mathcal{J}$ is *identical* to our $\mathcal{L}$. Ultimately, Garg et al. (2021) draw different conclusions from this saddle-point formulation, and propose to solve it by computing $\pi_Q = \arg\min_\pi \mathcal{J}(Q)$ and optimize the *dual function* $g(Q) = \min_\pi \mathcal{L}(\pi; Q)$. This function, however, can be highly nonsmooth and difficult to optimize, which is why `IQ-Learn` needs to heavily rely on regularization both in $\pi$ and $Q$. In contrast, our algorithm can be seen as trying to optimize the *primal function* $f(\pi) = \max_Q \mathcal{L}(\pi; Q)$ in terms of the policy $\pi$, which can be done in a stable way by incremental policy updates. Additionally, as Proposition 1 clearly reveals, optimizing the primal objective allows us to directly reason about the performance of the output policy. In contrast, we do not see a clear way to do this for the dual objective optimized by `IQ-Learn`.

Furthermore, we also note that `SPOIL` shares similarities with the algorithm `AdVIL` proposed by Swamy et al. (2021). Specifically, both `SPOIL` and `AdVIL` consider the same objective $\mathcal{L}$ but the two methods differ in their proposed algorithmic solutions and analytical approaches. Notably, Swamy et al. (2021) employed simultaneous gradient descent-ascent updates that made little use of the specific problem structure, whereas we consider an asymmetric scheme where the policy player uses mirror descent and the $Q$-player plays the best response. Therefore, our approach is more akin to minimizing the function $\pi \mapsto \max_{Q \in \mathcal{Q}} \mathcal{L}(\pi, Q)$ rather than using a primal-dual scheme. This is an important difference since Proposition 1 makes evident that the best response update of the $Q$ player is crucial for our analysis.

In what follows, we instantiate `SPOIL` in two settings of particular interest, depending on the Q-function class being used. We first provide a set of results for linear function approximation (where the algorithm is easy to implement and analyze) and for general function classes (where implementation and analysis are both less straightforward). We also discuss the convex case in Appendix B.8.

### 3.1 `SPOIL` for linear function approximation

We first provide a set of guarantees under the assumption that the function class is linear in some known features that realize the action-value functions of all softmax linear policies $\pi$ as linear combinations (see Assumption 1). In this setting, the actor and critic updates both simplify. For the actor, notice that the policy update can be rewritten as $\pi_k(a \mid x) \propto e^{\eta \sum_{i=1}^{k-1} Q_i(x,a)}$, which only requires storing $\sum_{i=1}^{k-1} Q_i$ in memory. For linear function approximation, this means that it suffices to maintain a single $d$-dimensional vector $\bar{\theta}_{k-1} = \sum_{i=1}^{k-1} \theta_i$ in memory and update it incrementally after each critic update. As for the critic update itself, notice that the objective function $\mathcal{L}$ and its empirical counterpart $\widehat{\mathcal{L}}$ can be rewritten in terms of the gap between the feature-expectation vectors

$$g_k = \mathbb{E}_{(X,A)\sim\mu^{\pi_{\mathrm{E}}}}[\varphi(X,A) - \varphi(X,\pi_k)], \quad \text{and} \quad \widehat{g}_k = \frac{1}{\tau_{\mathrm{E}}} \sum_{i=1}^{\tau_{\mathrm{E}}} \left(\varphi\left(X_{\mathrm{E}}^i, A_{\mathrm{E}}^i\right) - \varphi\left(X_{\mathrm{E}}^i, \pi_k\right)\right).$$

When considering linear functions $Q_\theta \colon (x, a) \mapsto \langle\varphi(x, a), \theta\rangle$, the objective can be written as

$$\mathcal{L}(\pi_k; Q_\theta) = \langle\theta, g_k\rangle, \quad \text{and} \quad \widehat{\mathcal{L}}(\pi_k; Q_\theta) = \langle\theta, \widehat{g}_k\rangle,$$

and the critic update can be simply written as $\theta_k = \arg\max_{\theta\in\mathfrak{B}(B_\theta)} \langle\theta, \widehat{g}_k\rangle$, which is trivial to compute. All in all, both actor and critic updates can be performed efficiently while only working in a $d$-dimensional Euclidean space. The following theorem provides our main result for `SPOIL`.

**Theorem 1.** *Let Assumption 1 hold. Run Algorithm 1 for $K = \frac{2\log A}{(1-\gamma)^2\varepsilon^2}$ iterations, with a learning rate $\eta = (1-\gamma)\sqrt{2\log A/K}$, and $\tau_{\mathrm{E}} = \mathcal{O}\left(\frac{d}{(1-\gamma)^2\varepsilon^2}\log\left(\frac{B_\theta B_\varphi A}{(1-\gamma)\varepsilon}\right)\right)$ samples collected by any expert policy $\pi_{\mathrm{E}}$. Then, the output satisfies $\mathbb{E}\left[\rho^{\pi_{\mathrm{E}}} - \rho^{\pi^{\mathrm{out}}}\right] = \mathcal{O}(\varepsilon)$.*

| **Algorithm 1** SPOIL with linear FA | **Algorithm 2** SPOIL with general FA |
|---|---|
| **Input:** Number of expert trajectories $\tau_\text{E}$, learning rate $\eta$, number of iterations $K$. | **Input:** Number of expert trajectories $\tau_\text{E}$, learning rate $\eta$, number of iterations $K$. |
| **Initialize:** $\theta_0 = 0$, uniform policy $\pi_0$. | **Initialize:** $Q_0 = 0$, uniform policy $\pi_0$. |
| **For** $k = 1, 2, \ldots, K$**:** | **For** $k = 1, 2, \ldots, K$**:** |
| 1. $\pi_k(a \mid x) \propto \pi_{k-1}(a \mid x) e^{\eta \langle \varphi(x,a), \theta_{k-1} \rangle}$. | 1. $\pi_k(a \mid x) \propto \pi_{k-1}(a \mid x) e^{\eta Q_{k-1}(x,a)}$. |
| 2. $\widehat{g}_k = \tau_\text{E}^{-1} \sum_{i=1}^{\tau_\text{E}} \big( \varphi \big( X_\text{E}^i, A_\text{E}^i \big) - \varphi \big( X_\text{E}^i, \pi_k \big) \big)$. | 2. $Q_k \in \underset{Q \in \mathcal{Q}}{\arg\max} \, \widehat{\mathcal{L}}(\pi_k, Q)$. |
| 3. $\theta_k = \underset{\theta : \|\theta\| \leq B_\theta}{\arg\max} \langle \theta, \widehat{g}_k \rangle = \dfrac{B_\theta}{\|\widehat{g}_k\|} \widehat{g}_k$. | **Output:** $\pi^\text{out} = \pi_I$, where $I \sim \mathcal{U}([K])$. |
| **Output:** $\pi^\text{out} = \pi_I$, where $I \sim \mathcal{U}([K])$. | |

The proof is in Appendix B.5. It is important to highlight that no assumptions are made concerning the expert policy. In particular, we do not require knowledge of a class $\Pi^\text{E}$ realizing the expert policy and as a consequence the bound on $\tau_\text{E}$ does not scale at all with a complexity measure of $\Pi^\text{E}$. This is in stark contrast with the theoretical guarantees for behavioural cloning (*e.g.*, Agarwal et al., 2022, Chapter 15, and Foster et al., 2024) which show bounds on the expert samples scaling with $\log |\Pi^\text{E}|$ (or the log covering number for continuous classes). It follows that no matter how complex the expert policy is, SPOIL suffers only the complexity of the environment (*i.e.*, the feature dimensionality $d$). Before moving to the next section, we emphasize that for consistency with the literature, Table 1 reports the number of expert trajectories required to guarantee that the difference between unnormalized returns, $(1 - \gamma)^{-1} \mathbb{E}\big[\rho^{\pi_\text{E}} - \rho^{\pi^\text{out}}\big]$, is bounded by $\mathcal{O}(\varepsilon)$.

### 3.2   SPOIL **for general function approximation**

For more complex $Q^\pi$-realizable MDPs, we analyze the version of SPOIL given in Algorithm 2. Notice that the updates can no longer use the linear structure of the value functions, and thus the critic update cannot be computed in closed form. Nevertheless, the algorithm remains well-defined, and satisfies the following performance guarantee.

**Theorem 2.** *Let Assumption 2 hold. Run Algorithm 2 for $K = \frac{2 \log A}{(1 - \gamma)^2 \varepsilon^2}$ iterations, with a learning rate $\eta = (1 - \gamma) \sqrt{2 \log A / K}$ and $\tau_\text{E} = \mathcal{O}\Big( \frac{\log A}{(1 - \gamma)^4 \varepsilon^4} \log \Big( \frac{\mathcal{N}_{\varepsilon'}(\mathcal{Q}, \|\cdot\|_\infty)}{\varepsilon(1 - \gamma)} \Big) \Big)$ samples collected by any expert $\pi_\text{E}$, where $\varepsilon' = \big( 8\sqrt{2} K^{3/2} A \log A \big)^{-1}$. Then, the output satisfies $\mathbb{E}\big[\rho^{\pi_\text{E}} - \rho^{\pi^\text{out}}\big] = \mathcal{O}(\varepsilon)$.*

There are two important remarks for the nonlinear extension. First, the maximization of $\widehat{\mathcal{L}}(\pi_k; Q)$ with respect to $Q$ is no longer available in closed form and it might not even be a concave optimization problem depending on the choice of the function class $\mathcal{Q}$. Therefore, computational efficiency cannot be ensured. Nevertheless, the form of the objective function remains very simple in terms of $Q$, and is arguably easier to optimize than other popular objective functions that are routinely optimized within deep RL with good empirical success (*e.g.*, the objective functions appearing in Mnih et al., 2015) and deep IL (Garg et al., 2021). Secondly, the expert sample complexity bound degrades from $\mathcal{O}(\varepsilon^{-2})$ achieved in the linear case to $\mathcal{O}(\varepsilon^{-4})$ in the nonlinear case due to the higher complexity of the policies produced by the algorithm (which results in a larger covering number of the policy class as highlighted in the proof sketch included in the next section).

## 4   Analysis

In this section we outline the proof of our two main results. Both proofs are based on two key steps which are self-evident from Proposition 1. The first one consists of a regret analysis to show that $\sum_{k=1}^{K} \mathcal{L}(\pi_k; Q_k)$ is bounded sublinearly in $K$. At a high level, the proof makes use of a classic technique of decomposing the "global" regret into the average of "local" regrets in each MDP state, first proposed by Even-Dar et al. (2004, 2009) and used in numerous other works (*e.g.*, Abbasi-Yadkori et al., 2019; Geist et al., 2019; Lan, 2023; Moulin and Neu, 2023). In proving this result,

a little care is needed in handling the potentially nonstationary nature of the expert policy. We circumvent the issue by using the performance difference lemma and controlling the regret at each state against the stationary comparator which induces the same state-action occupancy measure of the expert. Formally, we have the following bound, which we prove in Appendix B.2.

**Lemma 2.** *For any $k$ and any state-action pair $(x, a)$, consider the sequence of policies starting with $\pi_1$ as the uniform policy and updated as $\pi_{k+1}(a \mid x) \propto \pi_k(a \mid x) e^{\eta Q_k(x,a)}$ for some function $Q_k \colon \mathcal{X} \times \mathcal{A} \to \mathbb{R}$ such that $\|Q_k\|_\infty \leq \frac{1}{1-\gamma}$. Then, $\sum_{k=1}^{K} \mathcal{L}(\pi_k; Q_k) \leq \frac{\log A}{\eta} + \frac{\eta K}{2(1-\gamma)^2}$.*

This lemma applies to both the linear and nonlinear settings. The next and final step of the analysis is to establish concentration of the empirical objective and bound $\Delta(\pi_k)$ for each $k$. The main challenge in this step is the correlation between the iterates $\{\pi_k\}_{k=1}^{K}$ and the expert dataset. This can be handled via a uniform bound over the policy class to which all the algorithm iterates belong to. Importantly, this class is much smaller than the class of all policies, and allows us to make massive sample-complexity savings as compared to methods that need to control estimation errors associated with arbitrary policies. We provide the technical details separately for the linear and nonlinear cases.

## 4.1 Linear function approximation

In order to bound the estimation errors $\Delta(\pi_k)$, we apply a covering argument over the class of linear softmax policies. We have the following result.

**Lemma 3.** *Let $\{\pi_k\}_{k \in [K]}$ be the sequence of policies generated by Algorithm 1 and let $\Delta(\pi_k)$ be defined as in Proposition 1. Then, with probability at least $1 - \delta$, it holds that for all $k \in [K]$*

$$\Delta(\pi_k) \leq \frac{1}{K} + 4\sqrt{\frac{d}{(1-\gamma)^2 \tau_{\mathrm{E}}} \log\left(\frac{2 + 32K^2 \eta B_\theta B_\varphi A}{(1-\gamma)\delta}\right)}.$$

We defer the proof to Appendix B.4. We can use the above result to sketch the proof of Theorem 1.

*Proof sketch of Theorem 1.* Using Lemma 2 with $\eta = (1-\gamma)\sqrt{\frac{2\log A}{K}}$ and dividing by $K$, we obtain that $\frac{1}{K}\sum_{k=1}^{K} \mathcal{L}(\pi_k; Q_k) \leq \sqrt{\frac{2\log A}{(1-\gamma)^2 K}}$. Therefore, setting $K = \frac{2\log A}{(1-\gamma)^2 \varepsilon^2}$ guarantees $\frac{1}{K}\sum_{k=1}^{K} \mathcal{L}(\pi_k; Q_k) \leq \varepsilon$. Then, using the high-probability bound in Lemma 3 and the fact that $K^{-1}\sum_{k=1}^{K} \Delta(\pi_k)$ is a random variable bounded by $2(1-\gamma)^{-1}$ almost surely, we obtain the following expectation bound which holds for all $\delta > 0$

$$\frac{1}{K}\sum_{k=1}^{K} \mathbb{E}[\Delta(\pi_k)] \leq \frac{1}{K} + C\sqrt{\frac{d}{(1-\gamma)^2 \tau_{\mathrm{E}}} \log\left(\frac{B_\theta B_\varphi A}{(1-\gamma)^3 \varepsilon^2 \delta}\right)} + \frac{2\delta}{1-\gamma},$$

for some $C \in \mathbb{R}$. Noticing that the choice of parameters ensures $\frac{1}{K} \leq \frac{\varepsilon}{2}$ and setting $\delta = \frac{\varepsilon(1-\gamma)}{4}$ and $\tau_{\mathrm{E}} \geq \frac{C^2 d}{(1-\gamma)^2 \varepsilon^2} \log\left(\frac{B_\theta B_\varphi A}{(1-\gamma)^3 \varepsilon^2 \delta}\right)$, this bound implies that $\frac{2}{K}\sum_{k=1}^{K} \mathbb{E}[\Delta_k] \leq 4\varepsilon$. Invoking Proposition 1, we conclude that $\mathbb{E}\left[\rho^{\pi_{\mathrm{E}}} - \rho^{\pi^{\mathrm{out}}}\right] \leq 5\epsilon$. The full proof is in Appendix B.5. $\qquad\square$

## 4.2 General function approximation

The proof for the nonlinear setup follows the same conceptual steps but requires a more general concentration result for the objective function. Namely, the following lemma is the general counterpart of Lemma 3. The feature dimension $d$ appearing in the linear case is replaced by the complexity (as measured by the covering number) of the policy and value function classes containing the iterates.

**Lemma 4.** *Let $\mathcal{Q} \subset \mathbb{R}^{\mathcal{X} \times \mathcal{A}}$ denote an arbitrary class, $\{\pi_k\}_{k=1}^{K}$ denote the iterates produced by Algorithm 2, and let $\Delta(\pi_k)$ be defined as in Proposition 1. Then, with probability at least $1 - \delta$, it holds that for all $k \in [K]$*

$$\Delta(\pi_k) \leq \frac{1}{K} + \sqrt{\frac{8(K+1)\log\left(2\mathcal{N}_{\frac{1-\gamma}{8K^2 \eta A}}(\mathcal{Q}, \|\cdot\|_\infty)/\delta\right)}{(1-\gamma)^2 \tau_{\mathrm{E}}}}.$$

The proof is in Appendix B.6. Note that in the general case, the complexity of the policy class can increase linearly with the number of iterations $K$ (see Lemma 7). On the contrary, in the linear case, the policies generated by Algorithm 1 are parameterized by $d$ parameters and only the magnitude of these parameters increases with $K$. With this lemma, we present the proof sketch of Theorem 2.

*Proof sketch of Theorem 2.* Applying the decomposition in Proposition 1, the regret bound in Lemma 2, the concentration in Lemma 4, we obtain $\mathbb{E}\left[\rho^{\pi_E} - \rho^{\pi^{out}}\right] = \tilde{\mathcal{O}}\left(\frac{1}{\sqrt{K}} + \sqrt{\frac{K}{\tau_E}}\right)$. Setting $K = \tilde{\mathcal{O}}\left(\varepsilon^{-2}\right)$, and $\tau_E = \tilde{\mathcal{O}}\left(\varepsilon^{-4}\right)$, we get $\mathbb{E}\left[\rho^{\pi_E} - \rho^{\pi^{out}}\right] = \varepsilon$. The full proof is in Appendix B.7. □

## 5 Numerical experiments

We conduct experiments to verify that we can efficiently imitate complex experts in linear $Q^\pi$ environments, and can achieve massive improvements over behavioral cloning with large policy classes.[1]

To investigate this, we consider a randomly generated large linear MDP (a special case of linear $Q^\pi$-realizable MDP) with $|\mathcal{X}| = 500$ and $A = 1000$ but with a small feature dimension $d = 7$. We instantiate two experts. The first expert is trained to be the optimal softmax linear policy in this environment. This policy is parametrized by only $d$ parameters and can be realized by the class of softmax linear policies defined in Assumption 1, denoted $\Pi_{\text{lin}}^E$ here. In addition, we consider the second expert, which belongs to the class of three-layer neural networks denoted by $\Pi_{\text{NN}}^E$. This expert was trained to minimize the KL divergence with respect to the linear expert. As evident from Figure 1, our algorithm SPOIL performs well for both experts. This is in perfect agreement with the theory which provides a sample complexity bound that is independent of the expert policy class. On the other hand, behavioural cloning (BC) struggles with the complexity of the neural network expert policy class, and performs much worse. This is despite the fact that the dataset perfectly satisfies the realizability condition required by BC. This clearly demonstrates that complex behavior policies may indeed be problematic for BC to deal with, and we expect that such issues may also cause real performance drops in practical applications as well. Notice that in this experiment, SPOIL outperforms BC because the environment complexity is much lower

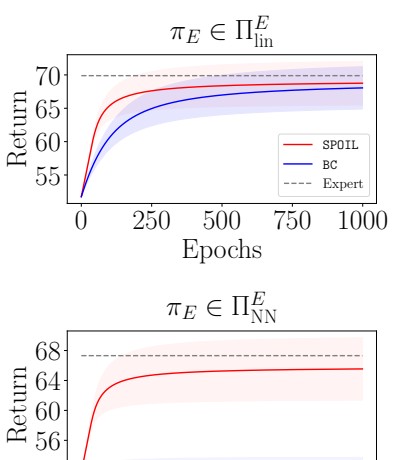

Figure 1: Experiments with simple and complex experts. Curves are averaged across 10 seeds.

than the policy class complexity. For fairness, we point out that the opposite situation is not unusual in RL and IL. In that case, it is reasonable to expect BC to be superior to SPOIL.

### 5.1 Continuous states experiments

We run the general function approximation version of our algorithm in continuous-states environments from the gym library (Towers et al., 2025). In particular, we consider the environments CartPole-v1, Acrobot-v1 and LunarLander-v2 where the expert is trained via Soft DQN. We use the expert data provided in the code base of Garg et al. (2021). The learner aims at imitating the expert performance given as input a variable number of expert trajectories. In order to make the task more challenging the trajectories are subsampled each 20 steps in CartPole-v1, Acrobot-v1 and each 5 in LunarLander-v2.[2] We compare the performance of the best policy found by each of these algorithms as a function of the number of expert trajectories given as input. In practice the maximization $\arg\max_{Q \in \mathcal{Q}} \widehat{\mathcal{L}}(\pi_k, Q)$ is approximated by performing a gradient ascent step. On the other hand, the actor update is approximated via Soft DQN (Haarnoja et al., 2017). In Figure 2, we

---

[1]Code is available at: https://github.com/antoine-moulin/spoil.

[2]This is common practice in IL experiments (see, *e.g.*, Garg et al., 2021).

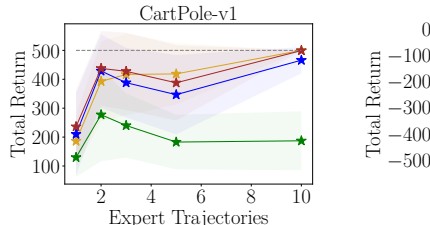
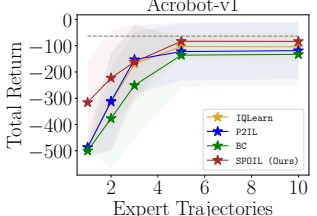
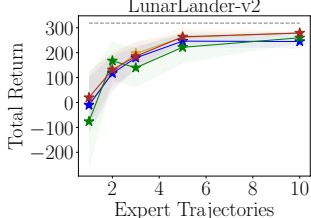

Figure 2: Experiments in continuous-state domains. Curves are averaged across 10 seeds.

can see that SPOIL performs comparably to the state-of-the-art algorithm IQ-Learn (Garg et al., 2021) and improves upon BC (Pomerleau, 1991; Foster et al., 2024) and P²IL (Viano et al., 2022).

## 6   Conclusions

In this work, we proposed analyses that leverage structural assumptions on the MDP without requiring trajectory access. This is made possible thanks to a novel regret decomposition that shifts the focus from updating a reward sequence based on expert data to updating a sequence of state-action value functions. To the best of our knowledge, these are the first rigorous theoretical guarantees for IL methods that learn state-action value functions from expert data, a technique popularized in practice by Garg et al. (2021). Among the many potential ways to extend and improve our work, we highlight two possible future directions below.

**Better rates in the general case.**   The most interesting immediate question that one can ask about our result is if the $\mathcal{O}(\varepsilon^{-4})$ scaling featured in our general bound is improvable under the conditions we assume. As a first step, we show an improvement for the case of convex class $\mathcal{Q}$ in Appendix B.8. However, we believe that substantially different algorithmic and analytic ideas would be necessary to answer this question for non convex classes, but we also think that our primal-dual framework provides a good starting point towards making such improvements. Furthermore, we would be curious to investigate appropriate notions of misspecification that our algorithm can deal with. It can be easily shown that requiring $Q^\pi$-realizability only up to a worst-case additive error of order $\varepsilon_{\mathrm{approx}}$ would incur the same additional term in the error bounds, but we believe that this assumption is too strong to warrant interest and we did not include an explicit statement. A much more interesting question is if this approximation guarantee would only be required to hold locally in the state-action pairs visited by the expert.

**Learning from features only.**   In the case of linear function approximation, the current approach critically relies on observing the expert state-action pairs to compute the vectors $\{\widehat{g}_k\}_{k=1}^K$. It would be interesting to check if an alternative algorithm can achieve the same guarantees by only observing the expert feature vectors instead. Another related direction is to efficiently imitate an expert from state-only trajectory given trajectory access to a linear-$Q^\pi$ realizable MDP.

Finally, let us remark that all previous theory work has focused either on imitation learning with additional trajectory access to the environment, both in tabular MDPs (Shani et al., 2022; Xu et al., 2023) and with additional structural assumptions (Liu et al., 2022; Viano et al., 2022, 2024; Moulin et al., 2025), or learning based on offline data only but under structural assumptions about the policy class used by the expert (Rajaraman et al., 2021; Swamy et al., 2022; Foster et al., 2024; Rohatgi et al., 2025). The first of these assumptions is clearly more restrictive than what we have considered in this work, and we have pointed out potential issues with the second set of methods when the policy class is exceedingly complex. This is not to say though that we consider our approach strictly superior to policy-based IL methods: as is often the case in RL, there is no single approach that dominates all others in all problems, and sometimes policy-based methods are more suitable for the job than value-based ones. Thus, even if our approach is not the ultimate answer to all questions in imitation learning, our results show that it is one potential alternative to consider in situations where other methods fail.

## Acknowledgments and Disclosure of Funding

The authors wish to thank Emmanuel Esposito for suggesting to look at the convex case and Akshay Krishnamurthy for an insightful discussion about our work. Luca Viano is funded through a PhD fellowship of the Swiss Data Science Center, a joint venture between EPFL and ETH Zurich. Gergely Neu and Antoine Moulin are funded via a European Research Council (ERC) project, under the European Union's Horizon 2020 research and innovation programme (Grant agreement No. 950180).

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

# Contents of Appendix

# A  Additional related works

Classical analyses by Ross and Bagnell (2010); Ross et al. (2011) on behavioural cloning (BC) established an error propagation framework relating the suboptimality of the learned policy to the worst-case generalization error incurred in predicting the expert policy. They proved that this suboptimality gap is upper-bounded by the generalization error up to a multiplicative factor $H^2$ (where $H$ is the horizon), a factor that is unavoidable when using the 0-1 loss for supervised learning. However, these results do not quantify the *expert sample complexity*, or the number of samples required to make the generalization error small.

A recent line of work has begun to investigate the expert sample complexity assuming knowledge of a policy class $\Pi^{\mathrm{E}}$ that realizes (or nearly realizes) the expert policy. For instance, Rajaraman et al. (2021) assume that the expert is deterministic and belongs to the class of deterministic linear policies $\Pi_{\mathrm{det,lin}}$ (defined in the caption of Table 1). They prove a bound on the required number of expert samples of order $\widetilde{\mathcal{O}}\big((H^2 d)/\varepsilon\big)$, where $d$ is the feature dimension in the definition of $\Pi_{\mathrm{det,lin}}$. Their technique is a reduction to the problem of multiclass classification in supervised learning, but their result is not informative for settings with general stochastic expert policies.

Further contributions to understanding the sample complexity of IL under policy class assumptions were made by Foster et al. (2024). Specifically, assuming the expert is included within a known class, $\pi_{\mathrm{E}} \in \Pi^{\mathrm{E}}$, they showed that one can learn an $\varepsilon$-optimal policy (as defined in Equation (2)) after observing $\mathcal{O}\big((H^2 \log|\Pi^{\mathrm{E}}|)/\varepsilon\big)$ samples for a deterministic expert or $\mathcal{O}\big((H^2 \log|\Pi^{\mathrm{E}}|)/\varepsilon^2\big)$ samples for a stochastic one (we report the dense reward case for brevity, though their bounds improve for sparse rewards). Addressing scenarios where the expert policy might only be almost well-specified, Rohatgi et al. (2025) demonstrate that there exists a computationally efficient algorithm that outputs an $\varepsilon$-optimal policy up to an additional approximation error of $H \log(W) \min_{\pi \in \Pi^{\mathrm{E}}} \mathcal{D}_{\mathsf{H}}^2(\mathbb{P}^\pi, \mathbb{P}^{\pi_{\mathrm{E}}})$. In this context, $\mathbb{P}^\pi$ is the trajectory distribution induced by $\pi$, $W$ is a density ratio defined as

$$W = \max_{\pi \in \Pi^{\mathrm{E}}} \max_{(x,a) \in \mathcal{X} \times \mathcal{A}} \max_{h \in [H]} \frac{\pi_{\mathrm{E},h}(a \mid x)}{\pi_h(a \mid x)} \,.$$

It is worth noting that these guarantees become vacuous when the policy class $\Pi^{\mathrm{E}}$ is such that at least one policy in $\Pi^{\mathrm{E}}$ fails to provide sufficient coverage for the expert's actions (leading to $W = +\infty$ as $\pi_h(a \mid x)$ gets close to zero for relevant state-action pairs and timestep where $\pi_{\mathrm{E},h}(a \mid x) > 0$), or if the minimum Hellinger distance $\min_{\pi \in \Pi^{\mathrm{E}}} \mathcal{D}_{\mathsf{H}}^2(\mathbb{P}^\pi, \mathbb{P}^{\pi_{\mathrm{E}}})$ is large. Alternatively, Foster et al. (2024) proved a misspecification result where the additional error is $\min_{\pi \in \Pi^{\mathrm{E}}} \chi^2(\mathbb{P}^\pi, \mathbb{P}^{\pi_{\mathrm{E}}})$. This misspecification error is measured by the $\chi^2$ divergence, with a leading coefficient constant in $H$ and $W$. However, the $\chi^2$ divergence is an upper bound on the Hellinger distance that is often way too loose to be practical. In a similar vein, Espinosa Dice et al. (2025) proved a benefit in terms of error propagation for a local search algorithm over behavioural cloning in misspecified settings, under the assumption that the learned policy is allowed to reset to states visited in the expert dataset.

Our work aligns with the recent renewed interest in proving refined expert sample complexity guarantees for offline imitation learning but distinguishes itself by swapping out the expert realizability assumption with a structural assumption on the environment. Early explorations for similar settings can be found in classical works by Abbeel and Ng (2004) and Syed and Schapire (2007). These studies proposed offline learning algorithms for MDPs with reward functions linear in a collection of features known to the learner, under the assumption that transition dynamics of the environment is also known. Versions of their approaches that do not assume such knowledge typically incur a worse sample complexity and often apply only in the tabular setting. Our work generalizes these classical approaches by removing the need for known transitions and for rewards to be linear in the features, as well as going beyond tabular MDPs. Notably, the linear $Q^\pi$-realizability assumption can hold even if the reward function and the transition dynamics are nonlinear. We summarize our comparison with these and other related works in Table 1.

Our work focuses on *learning a Q-value from expert data* and, in this regard, is closely related to the practical work of Garg et al. (2021). The novel regret decomposition employed in our analysis of SPOIL demonstrates, we believe for the first time, that provable guarantees are achievable by directly learning an action-value function from expert data. This contrasts with the majority of theoretical and practical imitation learning approaches, which typically first use the expert data to learn a reward function and subsequently use this learned reward function to infer an action-value function.

As we mentioned, SPOIL is very related to AdVIL. However, a key difference lies in the analysis: Swamy et al. (2021) conduct an error propagation analysis for AdVIL. From this, they conclude that AdVIL is equivalent to BC in the sense that if the loss for either method is at most $\varepsilon$ in every state, then the suboptimality of the extracted policy in an episodic setting with horizon $H$ is of order $H^2\varepsilon$ for both. However, this type of result does not investigate the assumptions or the number of samples needed to ensure these losses are indeed less than $\varepsilon$. Our work addresses this open question, establishing a clear distinction between the sample complexities of SPOIL and BC. Specifically, SPOIL and BC (and their respective analyses) rely on largely orthogonal sets of assumptions, making the two approaches complementary to each other: we expect SPOIL to be more suitable for imitation tasks with complex experts but simpler environments, while BC may be the preferred choice when this situation is reversed. Our sample complexity analysis for SPOIL critically relies on the $Q$-player using a best response strategy, and it is unlikely that equivalent results could be achieved using a standard gradient ascent step for the $Q$-player instead.

Very recently, Simchowitz et al. (2025) analyzed the error propagation properties of offline imitation learning algorithms in continuous action MDPs, showing that an exponential dependence on the horizon of the problem is unavoidable if no structure is imposed on the environment. On the other hand, the same authors point out that if the state-action value functions were Lipschitz in the action space, then efficient learning would be possible. Conceptually, we believe that the SPOIL algorithm could also be applied in the continuous action case. Such an extension would suggest that another scenario enabling effective imitation learning in continuous action spaces arises when the learner has access to a suitably expressive class of state-action value functions.

Following a similar line of research that studies imitation learning from a control-theoretic perspective, Block et al. (2023) studied guarantees for generative behavioural cloning, assuming access to a stabilizing policy dubbed a *synthesis oracle*. These policies can be computed exactly if the dynamics are known, an assumption which is not imposed in our work. However, when provided with such an oracle, Block et al. (2023) derive bounds on a stricter metric for imitation. Specifically, they bound the probability that expert and learner trajectories diverge at some time step, as opposed to the difference in cumulative return that we analyze in our work.

# B  Omitted proofs

In this appendix, we provide the omitted proofs of the main results.

## B.1  Proof of Lemma 1 (performance difference lemma)

We start presenting the performance difference lemma proven in a more general form which allows one policy to be nonstationary.

**Lemma 1.** *Let $\pi$ be a stationary policy and $\pi'$ be any policy. Then,*

$$\rho^{\pi'} - \rho^{\pi} = \mathbb{E}_{(X,A)\sim\mu^{\pi'}}[Q^{\pi}(X,A) - V^{\pi}(X)] \,.$$

*Proof.* Consider the Bellman equations for the stationary policy $\pi$. For any state-action pair $(x,a)$, we have

$$Q^{\pi}(x,a) = r(x,a) + \gamma \sum_{x'\in\mathcal{X}} P(x' \,|\, x,a)V^{\pi}(x') \,.$$

Averaging both sides with the distribution $\mu^{\pi'}$ and reordering the terms, we obtain

$$\sum_{x,a} \mu^{\pi'}(x,a)r(x,a) = \sum_{x,a} \mu^{\pi'}(x,a)\left(Q^{\pi}(x,a) - \gamma \sum_{x'\in\mathcal{X}} P(x' \,|\, x,a)V^{\pi}(x')\right)$$

$$= (1-\gamma)\sum_x \nu_0(x)V^{\pi}(x) + \sum_{x,a} \mu^{\pi'}(x,a)\big(Q^{\pi}(x,a) - V^{\pi}(x)\big) \,,$$

where we used the flow condition of the occupancy measure $\mu^{\pi'}$ in the last step (see Equation 1). The claim then follows by noticing that $\rho^{\pi} = (1-\gamma)\sum_x \nu_0(x)V^{\pi}(x)$ and $\rho^{\pi'} = \sum_{x,a} \mu^{\pi'}(x,a)r(x,a)$. $\qquad\square$

## B.2  Proof of Lemma 2 (regret of the policy player)

Next, we apply Lemma 14 to the special case of the exponential weights update, where the divergence is chosen to be the KL divergence, and use it to derive a bound on the regret of the policy player.

**Lemma 2.** *For any $k$ and any state-action pair $(x,a)$, consider the sequence of policies starting with $\pi_1$ as the uniform policy and updated as $\pi_{k+1}(a \,|\, x) \propto \pi_k(a \,|\, x)e^{\eta Q_k(x,a)}$ for some function $Q_k \colon \mathcal{X} \times \mathcal{A} \to \mathbb{R}$ such that $\|Q_k\|_\infty \le \frac{1}{1-\gamma}$. Then, $\sum_{k=1}^{K} \mathcal{L}(\pi_k; Q_k) \le \frac{\log A}{\eta} + \frac{\eta K}{2(1-\gamma)^2}$.*

*Proof.* Let us recall that

$$\mathcal{L}(\pi_k, Q_k) = \mathbb{E}_{(X,A)\sim\mu^{\pi_{\mathrm{E}}}}[Q_k(X,A) - Q_k(X,\pi_k)] \,,$$

where $\pi_{\mathrm{E}}$ is a potentially nonstationary policy. To continue, let us consider the stationary policy $\bar\pi_{\mathrm{E}} \colon \mathcal{X} \to \Delta(\mathcal{A})$ that induces the same state-action occupancy measure of the expert, *i.e.*, such that $\mu^{\bar\pi_{\mathrm{E}}} = \mu^{\pi_{\mathrm{E}}}$. This equality can be guaranteed by choosing, for any $(x,a) \in \mathcal{X} \times \mathcal{A}$, $\bar\pi_{\mathrm{E}}(a \,|\, x) = \frac{\mu^{\pi_{\mathrm{E}}}(x,a)}{\nu^{\pi_{\mathrm{E}}}(x)}$ if $\nu^{\pi_{\mathrm{E}}}(x) \ne 0$ and $\pi_0(a)$ otherwise, where $\pi_0 \in \Delta(\mathcal{A})$ is an arbitrary distribution. Then, we continue as follows

$$\mathcal{L}(\pi_k, Q_k) = \mathbb{E}_{(X,A)\sim\mu^{\pi_{\mathrm{E}}}}[Q_k(X,A) - Q_k(X,\pi_k)]$$

$$= \mathbb{E}_{(X,A)\sim\mu^{\bar\pi_{\mathrm{E}}}}[Q_k(X,A) - Q_k(X,\pi_k)]$$

$$= \sum_{x\in\mathcal{X}} \nu^{\bar\pi_{\mathrm{E}}}(x) \sum_{a\in\mathcal{A}} Q_k(x,a)(\bar\pi_{\mathrm{E}}(a \,|\, x) - \pi_k(a \,|\, x)) \,.$$

Summing over $k \in [K]$, we obtain

$$\sum_{k=1}^{K} \mathcal{L}(\pi_k, Q_k) = \sum_{x\in\mathcal{X}} \nu^{\bar\pi_{\mathrm{E}}}(x) \sum_{k=1}^{K} \sum_{a\in\mathcal{A}} Q_k(x,a)(\bar\pi_{\mathrm{E}}(a \,|\, x) - \pi_k(a \,|\, x)) \,.$$

It remains to prove the following bound.

$$\sum_{k=1}^{K} \sum_{a \in \mathcal{A}} Q_k(x,a)(\bar{\pi}_{\mathrm{E}}(a \mid x) - \pi_k(a \mid x)) \leq \frac{\log A}{\eta} + \frac{\eta K}{2(1-\gamma)^2} \,.$$

The result is proven as a particular case of Lemma 14. Specifically, we have that when $V$ is the $A$-dimensional simplex and the Bregman divergence is the KL divergence, it holds that

$$x_{k+1} = \underset{v \in V}{\arg\min} \left\{ \langle \ell_k, v \rangle + \frac{1}{\eta} D(v, x_k) \right\} = \frac{x_k \odot \exp(-\eta \ell_k)}{\langle \mathbf{1}, x_k \odot \exp(-\eta \ell_k) \rangle} \,,$$

where $\odot$ is the elementwise product. We apply Lemma 14 for each state $x \in \mathcal{X}$, replacing $x_k = \pi_k(\cdot \mid x)$ and $\ell_k = -Q_k(x, \cdot)$. We obtain that for the update $\pi_{k+1}(a \mid x) \propto \pi_k(a \mid x) e^{\eta Q_k(x,a)}$, the guarantee in Lemma 14 holds. Moreover, in this setting we have $\lambda = 1$, and $\ell_{\max} = \frac{1}{1-\gamma}$. Given that for any state-action pair $(x,a)$, the initial policy is $\pi_1(a \mid x) = \frac{1}{A}$, we have that $D(\pi(\cdot \mid x), \pi_1(\cdot \mid x)) \leq \log A$. Thus, we have the following bound

$$\sum_{a \in \mathcal{A}} Q_k(x,a)(\bar{\pi}_{\mathrm{E}}(a \mid x) - \pi_k(a \mid x)) \leq \frac{\log A}{\eta} + \frac{\eta K}{2(1-\gamma)^2} \,,$$

and the conclusion follows from $\nu^{\bar{\pi}_{\mathrm{E}}}$ being a probability distribution. $\square$

### B.3 General concentration argument

To prove the main results of this paper, we prove a general concentration inequality that we will use for the iterates produced by both Algorithm 1 and Algorithm 2. Specifically, when analyzing Algorithm 1, we consider the policy class $\Pi_{\mathrm{lin}}$ defined as follows

$$\Pi_{\mathrm{lin}} = \left\{ \pi \in \Delta(\mathcal{A})^{\mathcal{X}} : \exists (\theta_k)_{k \in [K]} \subset \mathfrak{B}(B_\theta), \pi(a \mid x) = \frac{\exp\left(\eta \sum_{k=1}^{K} \langle \varphi(x,a), \theta_k \rangle\right)}{\sum_{b \in \mathcal{A}} \exp\left(\eta \sum_{k=1}^{K} \langle \varphi(x,b), \theta_k \rangle\right)} \right\}, \tag{3}$$

while in the nonlinear case (Algorithm 2), we will consider the policy class

$$\Pi_{\mathcal{Q}} = \left\{ \pi \in \Delta(\mathcal{A})^{\mathcal{X}} : \exists (Q_k)_{k \in [K]} \subset \mathcal{Q}, \pi(a \mid x) = \frac{\exp\left(\eta \sum_{k=1}^{K} Q_k(x,a)\right)}{\sum_{b \in \mathcal{A}} \exp\left(\eta \sum_{k=1}^{K} Q_k(x,b)\right)} \right\}. \tag{4}$$

The result is the following.

**Lemma 5.** *Let $\mathcal{Q} \subset \mathbb{R}^{\mathcal{X} \times \mathcal{A}}$ be a value function class such that for any $Q \in \mathcal{Q}$, $\|Q\|_\infty \leq \frac{1}{1-\gamma}$. Consider the sequences of estimated objective functions $\{\widehat{\mathcal{L}}(\pi_k, \cdot)\}_{k=1}^{K}$ for a policy sequence $\{\pi_k\}_{k=1}^{K}$ belonging to a policy class $\Pi$. For any $k \in [K]$, recall that for any policy $\pi$ and function $Q$, the objective function is defined as*

$$\mathcal{L}(\pi; Q) = \mathbb{E}_{(X,A) \sim \mu^{\pi_{\mathrm{E}}}}[Q(X,A) - Q(X,\pi)].$$

*Then, with probability larger than $1 - \delta$, it holds that for all $k \in [K]$ simultaneously that*

$$\Delta(\pi_k) = \sup_{Q \in \mathcal{Q}} \left| \widehat{\mathcal{L}}(\pi_k, Q) - \mathcal{L}(\pi_k, Q) \right| \leq \inf_{\epsilon : \epsilon > 0} \left\{ \frac{4\epsilon}{1-\gamma} + \sqrt{\frac{8 \log\left(2 \mathcal{N}_\epsilon\left(\mathcal{Q} \times \Pi, \|\cdot\|_{\infty,1}\right)/\delta\right)}{(1-\gamma)^2 \tau_{\mathrm{E}}}} \right\},$$

*where, for any $(Q, \pi), (Q', \pi') \in \mathcal{Q} \times \Pi$, we defined the distance $\|(Q, \pi) - (Q', \pi')\|_{\infty,1} = \|Q - Q'\|_\infty + \max_{x \in \mathcal{X}} \|\pi(\cdot \mid x) - \pi'(\cdot \mid x)\|_1$.*

*Proof.* Let us recall that for any $Q \in \mathcal{Q}$ and any $k \in [K]$, we have

$$\widehat{\mathcal{L}}(\pi_k, Q) = \frac{1}{\tau_{\mathrm{E}}} \sum_{i=1}^{\tau_{\mathrm{E}}} \left( Q(X_{\mathrm{E}}^i, A_{\mathrm{E}}^i) - \sum_{a \in \mathcal{A}} \pi_k(a \mid X_{\mathrm{E}}^i) Q(X_{\mathrm{E}}^i, a) \right),$$

and notice that $\widehat{\mathcal{L}}(\pi_k, Q)$ is not an unbiased estimator of $\mathcal{L}(\pi_k, Q)$ since the policy $\pi_k$ depends on the expert data. Therefore, we aim at establishing a uniform concentration bound over the policy class $\Pi$. To this end, let us consider a fixed pair $(Q, \pi) \in \mathcal{C}_\epsilon(\mathcal{Q} \times \Pi, \|\cdot\|_{\infty,1})$, and notice that $\widehat{\mathcal{L}}(\pi, Q)$ is an average of random variables of the form

$$W_i = Q(X_{\mathrm{E}}^i, A_{\mathrm{E}}^i) - \sum_{a \in \mathcal{A}} \pi(a \mid X_{\mathrm{E}}^i) Q(X_{\mathrm{E}}^i, a) \,,$$

where $i \in [\tau_{\mathrm{E}}]$. Each $W_i$ is an unbiased estimator of $\mathcal{L}(\pi, Q)$ since $\pi$ is fixed (*i.e.*, $\pi$ is not a random quantity depending on the expert data) and $(X_{\mathrm{E}}^i, A_{\mathrm{E}}^i) \sim \mu^{\pi_{\mathrm{E}}}$ for all $i \in [\tau_{\mathrm{E}}]$. Thus, for any $i \in [\tau_{\mathrm{E}}]$, $\mathbb{E}[W_i] = \mathcal{L}(\pi, Q)$. Moreover, notice that for all $i \in [\tau_{\mathrm{E}}]$, $-\frac{2}{1-\gamma} \leq W_i \leq \frac{2}{1-\gamma}$. Therefore, by an application of Hoeffding's inequality (see Lemma 13), we have that for all $t > 0$,

$$\mathbb{P}\Big[\big|\widehat{\mathcal{L}}(\pi, Q) - \mathcal{L}(\pi, Q)\big| \geq t\Big] \leq 2 \exp\left(-\frac{t^2 \tau_{\mathrm{E}} (1-\gamma)^2}{8}\right).$$

That is, choosing $t = \frac{8 \log(2/\delta)}{(1-\gamma)^2 \tau_{\mathrm{E}}}$ guarantees that with probability at least $1 - \delta$,

$$\big|\widehat{\mathcal{L}}(\pi, Q) - \mathcal{L}(\pi, Q)\big| \leq \sqrt{\frac{8 \log(2/\delta)}{(1-\gamma)^2 \tau_{\mathrm{E}}}} \,.$$

Applying a union bound, we further have that with probability at least $1 - \delta$, for all $(Q, \pi) \in \mathcal{C}_\epsilon(\mathcal{Q} \times \Pi, \|\cdot\|_{\infty,1})$ it holds that

$$\big|\widehat{\mathcal{L}}(\pi, Q) - \mathcal{L}(\pi, Q)\big| \leq \sqrt{\frac{8 \log\Big(2\mathcal{N}_\epsilon\big(\mathcal{Q} \times \Pi, \|\cdot\|_{\infty,1}\big)/\delta\Big)}{(1-\gamma)^2 \tau_{\mathrm{E}}}} \,.$$

Recall that $\mathcal{C}_\epsilon(\mathcal{Q} \times \Pi, \|\cdot\|_{\infty,1})$ is assumed to be an $\epsilon$-covering set of the space $\mathcal{Q} \times \Pi$ with respect to the distanec $\|\cdot\|_{\infty,1}$. For any pair $(Q, \pi_k) \in \mathcal{Q} \times \Pi$, let $(Q_\epsilon, \pi_{k,\epsilon}) \in \mathcal{C}_\epsilon(\mathcal{Q} \times \Pi, \|\cdot\|_{\infty,1})$ denote the element of the covering such that $\|(Q, \pi_k) - (Q_\epsilon, \pi_{k,\epsilon})\|_{\infty,1} \leq \epsilon$. Then, we have that

$$\left|\widehat{\mathcal{L}}(\pi_k, Q) - \widehat{\mathcal{L}}(\pi_{k,\epsilon}, Q_\epsilon)\right| \leq \left|\frac{1}{\tau_{\mathrm{E}}} \sum_{i=1}^{\tau_{\mathrm{E}}} \big(Q(X_{\mathrm{E}}^i, A_{\mathrm{E}}^i) - Q_\epsilon(X_{\mathrm{E}}^i, A_{\mathrm{E}}^i)\big)\right|$$

$$+ \left|\frac{1}{\tau_{\mathrm{E}}} \sum_{i=1}^{\tau_{\mathrm{E}}} \sum_{a \in \mathcal{A}} \big(\pi_{k,\epsilon}(a \mid X_{\mathrm{E}}^i) Q_\epsilon(X_{\mathrm{E}}^i, a) - \pi_k(a \mid X_{\mathrm{E}}^i) Q(X_{\mathrm{E}}^i, a)\big)\right|$$

$$\leq \|Q - Q_\epsilon\|_\infty + \left|\frac{1}{\tau_{\mathrm{E}}} \sum_{i=1}^{\tau_{\mathrm{E}}} \sum_{a \in \mathcal{A}} \big(\pi_{k,\epsilon}(a \mid X_{\mathrm{E}}^i) - \pi_k(a \mid X_{\mathrm{E}}^i)\big) Q_\epsilon(X_{\mathrm{E}}^i, a)\right|$$

$$+ \left|\frac{1}{\tau_{\mathrm{E}}} \sum_{i=1}^{\tau_{\mathrm{E}}} \sum_{a \in \mathcal{A}} \pi_k(a \mid X_{\mathrm{E}}^i) \big(Q(X_{\mathrm{E}}^i, a) - Q_\epsilon(X_{\mathrm{E}}^i, a)\big)\right|.$$

Noting that for any $Q \in \mathcal{Q}$, $\|Q\|_\infty \leq \frac{1}{1-\gamma}$, and that for any state $x$, $\pi_k(\cdot \mid x) \in \Delta(\mathcal{A})$, using Hölder's inequality, we further have

$$\left|\widehat{\mathcal{L}}(\pi_k, Q) - \widehat{\mathcal{L}}(\pi_{k,\epsilon}, Q_\epsilon)\right| \leq \|Q - Q_\epsilon\|_\infty + \frac{\max_{x \in \mathcal{X}} \|\pi_{k,\epsilon}(\cdot \mid x) - \pi_k(\cdot \mid x)\|_1}{1-\gamma} + \|Q - Q_\epsilon\|_\infty$$

$$\leq \frac{2 \|(Q, \pi_k) - (Q_\epsilon, \pi_{k,\epsilon})\|_{\infty,1}}{1-\gamma}$$

$$\leq \frac{2\epsilon}{1-\gamma} \,,$$

where we used the definition of $(\pi_{k,\epsilon}, Q_\epsilon)$ and $\gamma \in (0,1)$ in the last inequality. Similarly, for the true objective we have that

$$
\begin{aligned}
|\mathcal{L}(\pi_k, Q) - \mathcal{L}(\pi_{k,\epsilon}, Q_\epsilon)| &\leq \left| \mathbb{E}_{(X,A) \sim \mu^{\pi_\mathrm{E}}} [Q(X,A) - Q_\epsilon(X,A)] \right| \\
&\quad + |\mathbb{E}_{X \sim \nu^{\pi_\mathrm{E}}} [Q(X,\pi_k) - Q_\epsilon(X,\pi_{k,\epsilon})]| \\
&\leq \|Q - Q_\epsilon\|_\infty + |\mathbb{E}_{X \sim \nu^{\pi_\mathrm{E}}} [Q(X,\pi_k) - Q(X,\pi_{k,\epsilon})]| \\
&\quad + |\mathbb{E}_{X \sim \nu^{\pi_\mathrm{E}}} [Q(X,\pi_{k,\epsilon}) - Q_\epsilon(X,\pi_{k,\epsilon})]| \\
&\leq \|Q - Q_\epsilon\|_\infty + \frac{\max_{x \in \mathcal{X}} \|\pi_{k,\epsilon}(\cdot \,|\, x) - \pi_k(\cdot \,|\, x)\|_1}{1 - \gamma} + \|Q - Q_\epsilon\|_\infty \\
&\leq \frac{2\epsilon}{1 - \gamma} \,.
\end{aligned}
$$

Therefore, with probability at least $1 - \delta$, it holds that for any $k \in [K]$ and any $Q \in \mathcal{Q}$,

$$
\begin{aligned}
\left| \widehat{\mathcal{L}}(\pi_k, Q) - \mathcal{L}(\pi_k, Q) \right| &\leq \left| \widehat{\mathcal{L}}(\pi_k, Q) - \widehat{\mathcal{L}}(\pi_{k,\epsilon}, Q_\epsilon) \right| + \left| \widehat{\mathcal{L}}(\pi_{k,\epsilon}, Q_\epsilon) - \mathcal{L}(\pi_{k,\epsilon}, Q_\epsilon) \right| \\
&\quad + |\mathcal{L}(\pi_k, Q) - \mathcal{L}(\pi_{k,\epsilon}, Q_\epsilon)| \\
&\leq \frac{4\epsilon}{1 - \gamma} + \sqrt{\frac{8 \log \left( 2 \mathcal{N}_\epsilon \left( \mathcal{Q} \times \Pi, \|\cdot\|_{\infty,1} \right) / \delta \right)}{(1 - \gamma)^2 \tau_\mathrm{E}}} \,.
\end{aligned}
$$

Moreover, since the above bound holds for all $Q \in \mathcal{Q}$, it holds for the supremum over this class. With probability at least $1 - \delta$, we have for any $k \in [K]$ that

$$
\sup_{Q \in \mathcal{Q}} \left| \widehat{\mathcal{L}}(\pi_k, Q) - \mathcal{L}(\pi_k, Q) \right| \leq \frac{4\epsilon}{1 - \gamma} + \sqrt{\frac{8 \log \left( 2 \mathcal{N}_\epsilon \left( \mathcal{Q} \times \Pi, \|\cdot\|_{\infty,1} \right) / \delta \right)}{(1 - \gamma)^2 \tau_\mathrm{E}}} \,.
$$

The proof is concluded by noting that the above proof holds for any covering size $\epsilon > 0$. $\qquad \square$

### B.4 Proof of Lemma 3 (concentration linear case)

We now instantiate Lemma 5 in the linear $Q^\pi$-realizable setting. For this purpose, we compute a bound on the covering number of the class $\Pi_\mathrm{lin}$, defined in Equation (3).

**Lemma 6** (Covering number of $\Pi_\mathrm{lin}$). *For $\epsilon > 0$, it holds that the $\epsilon$-covering number of the policy class $\Pi_\mathrm{lin}$ can be bounded as*

$$
\mathcal{N}_\epsilon(\Pi_\mathrm{lin}, \|\cdot\|_1) \leq \left( 1 + \frac{2K \eta B_\theta B_\varphi A}{\epsilon} \right)^d \,,
$$

*where, with a slight abuse of notation, $\|\cdot\|_1$ denotes the distance defined for any $\pi, \pi' \in \Pi_\mathrm{lin}$ as $\|\pi - \pi'\|_1 = \sup_{x \in \mathcal{X}} \|\pi(\cdot \,|\, x) - \pi'(\cdot \,|\, x)\|_1$. Moreover, let*

$$
\mathcal{Q}_\mathrm{lin} = \{ Q \colon \mathcal{X} \times \mathcal{A} \to \mathbb{R} : \exists \theta \in \mathfrak{B}(B_\theta), \forall (x,a) \in \mathcal{X} \times \mathcal{A}, \ Q(x,a) = \langle \theta, \varphi(x,a) \rangle \}
$$

*be the class of linear action-value functions. Then, it holds that*

$$
\mathcal{N}_\epsilon \left( \mathcal{Q}_\mathrm{lin} \times \Pi_\mathrm{lin}, \|\cdot\|_{\infty,1} \right) \leq \left( 1 + \frac{4K \eta B_\theta B_\varphi A}{\epsilon} \right)^{2d} \,.
$$

*Proof.* Let us consider two policies $\pi$ and $\pi'$ in the class $\Pi_\mathrm{lin}$. There exist $\theta_1, \ldots, \theta_K \in \mathfrak{B}(B_\theta)$ and $\theta'_1, \ldots, \theta'_K \in \mathfrak{B}(B_\theta)$ such that for any state-action pair $(x,a) \in \mathcal{X} \times \mathcal{A}$, $\pi$ and $\pi'$ can be written as

$$
\pi(a \,|\, x) = \frac{\exp \left( \eta \left\langle \varphi(x,a), \sum_{k=1}^K \theta_k \right\rangle \right)}{\sum_{b \in \mathcal{A}} \exp \left( \eta \left\langle \varphi(x,b), \sum_{k=1}^K \theta_k \right\rangle \right)} \,,
$$

and

$$
\pi'(a \,|\, x) = \frac{\exp \left( \eta \left\langle \varphi(x,a), \sum_{k=1}^K \theta'_k \right\rangle \right)}{\sum_{b \in \mathcal{A}} \exp \left( \eta \left\langle \varphi(x,b), \sum_{k=1}^K \theta'_k \right\rangle \right)} \,.
$$

In particular, let us fix a state $x \in \mathcal{X}$, and denote $\bar{\theta}_K = \sum_{k=1}^{K} \theta_k$, $\bar{\theta}'_K = \sum_{k=1}^{K} \theta'_k$. First, by Cauchy-Schwarz's inequality, we have

$$\|\pi(\cdot \,|\, x) - \pi'(\cdot \,|\, x)\|_1 \leq \sqrt{A} \,\|\pi(\cdot \,|\, x) - \pi'(\cdot \,|\, x)\| \,.$$

By 1-Lipschitzness of the softmax function (Lemma 15), it holds that

$$\begin{aligned}
\|\pi(\cdot \,|\, x) - \pi'(\cdot \,|\, x)\|_1 &\leq \eta \sqrt{A} \,\|\langle \varphi(x, \cdot), \bar{\theta}_K - \bar{\theta}'_K \rangle\| \\
&= \eta \sqrt{A \sum_{a \in \mathcal{A}} \left( \langle \varphi(x, a), \bar{\theta}_K - \bar{\theta}'_K \rangle \right)^2} \\
&\leq \eta \sqrt{A \sum_{a \in \mathcal{A}} \|\varphi(x, a)\|^2 \,\|\bar{\theta}_K - \bar{\theta}'_K\|^2} \qquad \text{(Cauchy-Schwarz)} \\
&\leq \eta B_\varphi A \,\|\bar{\theta}_K - \bar{\theta}'_K\| \,,
\end{aligned}$$

where the last inequality follows from the bound on the features $\varphi$ in Assumption 1. Notice that $\bar{\theta}_K, \bar{\theta}'_K \in \mathfrak{B}(KB_\theta)$. Therefore, the $\epsilon$-covering number for $\Pi_{\mathrm{lin}}$ with respect to the distance $\|\cdot\|_1$, $\mathcal{N}_\epsilon(\Pi_{\mathrm{lin}}, \|\cdot\|_1)$, is upper-bounded by the $\frac{\epsilon}{\eta B_\varphi A}$-covering number of the Euclidean ball $\mathfrak{B}(KB_\theta)$ with respect to the distance $\|\cdot\|$, and

$$\begin{aligned}
\mathcal{N}_\epsilon(\Pi_{\mathrm{lin}}, \|\cdot\|_1) &\leq \mathcal{N}_{\frac{\epsilon}{\eta B_\varphi A}}(\mathfrak{B}(KB_\theta), \|\cdot\|) \\
&\leq \left( 1 + \frac{2K\eta B_\theta B_\varphi A}{\epsilon} \right)^d,
\end{aligned}$$

where we used Lemma 16 in the last inequality. For the second part of the lemma, let us consider $Q, Q' \in \mathcal{Q}_{\mathrm{lin}}$. By definition of $\mathcal{Q}_{\mathrm{lin}}$, there exists $\theta, \theta' \in \mathfrak{B}(B_\theta)$ such that for any state-action pair $(x, a)$, $Q(x, a) = \langle \varphi(x, a), \theta \rangle$ and $Q'(x, a) = \langle \varphi(x, a), \theta' \rangle$. Then,

$$\max_{x, a \in \mathcal{X} \times \mathcal{A}} |Q(x, a) - Q'(x, a)| = \max_{x, a \in \mathcal{X} \times \mathcal{A}} |\langle \varphi(x, a), \theta - \theta' \rangle| \leq B_\varphi \,\|\theta - \theta'\| \,.$$

Therefore, the $\epsilon$-covering number of $\mathcal{Q}_{\mathrm{lin}}$, $\mathcal{N}_\epsilon(\mathcal{Q}_{\mathrm{lin}}, \|\cdot\|_\infty)$, is upper-bounded by the $\epsilon/B_\varphi$-covering number of the $d$-dimensional ball with radius $B_\theta$, $\mathcal{N}_{\epsilon/B_\varphi}(\mathfrak{B}(B_\theta), \|\cdot\|)$. We have

$$\mathcal{N}_\epsilon(\mathcal{Q}_{\mathrm{lin}}, \|\cdot\|_\infty) \leq \mathcal{N}_{\epsilon/B_\varphi}(\mathfrak{B}(B_\theta), \|\cdot\|) \leq \left( 1 + \frac{2B_\theta B_\varphi}{\epsilon} \right)^d \,.$$

Finally, the proof is concluded by noting that

$$\mathcal{N}_\epsilon\left( \mathcal{Q}_{\mathrm{lin}} \times \Pi_{\mathrm{lin}}, \|\cdot\|_{\infty, 1} \right) \leq \mathcal{N}_{\epsilon/2}(\Pi_{\mathrm{lin}}, \|\cdot\|_1) \mathcal{N}_{\epsilon/2}(\mathcal{Q}_{\mathrm{lin}}, \|\cdot\|_\infty) \,.$$

$\square$

Finally, the following result proves the concentration of the estimators used in Algorithm 1.

**Lemma 3.** *Let* $\{\pi_k\}_{k \in [K]}$ *be the sequence of policies generated by Algorithm 1 and let* $\Delta(\pi_k)$ *be defined as in Proposition 1. Then, with probability at least* $1 - \delta$, *it holds that for all* $k \in [K]$

$$\Delta(\pi_k) \leq \frac{1}{K} + 4 \sqrt{\frac{d}{(1-\gamma)^2 \tau_{\mathrm{E}}} \log\left( \frac{2 + 32K^2 \eta B_\theta B_\varphi A}{(1-\gamma)\delta} \right)} \,.$$

*Proof.* By Lemma 5, it holds that for all $k \in [K]$

$$\Delta(\pi_k) \leq \inf_{\epsilon:\epsilon>0}\left\{ \frac{4\epsilon}{1-\gamma} + \sqrt{\frac{8\log\left(2\mathcal{N}_\epsilon\left(\mathcal{Q}_{\text{lin}} \times \Pi_{\text{lin}}, \|\cdot\|_{\infty,1}\right)/\delta\right)}{(1-\gamma)^2\tau_{\text{E}}}} \right\}$$

$$\leq \frac{1}{K} + 2\sqrt{\frac{2\log\left(2\mathcal{N}_{(1-\gamma)/4K}\left(\mathcal{Q}_{\text{lin}} \times \Pi_{\text{lin}}, \|\cdot\|_{\infty,1}\right)/\delta\right)}{(1-\gamma)^2\tau_{\text{E}}}}$$

$$\leq \frac{1}{K} + 2\sqrt{\frac{2}{(1-\gamma)^2\tau_{\text{E}}}\log\left(\frac{2}{\delta}\left(1 + \frac{16K^2\eta B_\theta B_\varphi A}{1-\gamma}\right)^{2d}\right)}$$

$$\leq \frac{1}{K} + 4\sqrt{\frac{d}{(1-\gamma)^2\tau_{\text{E}}}\log\left(\frac{2 + 32K^2\eta B_\theta B_\varphi A}{(1-\gamma)\delta}\right)},$$

where the third inequality follows from Lemma 6. $\qquad\square$

## B.5 Proof of Theorem 1 (sample complexity guarantee for linear $Q^\pi$-realizable MDPs)

**Theorem 1.** *Let Assumption 1 hold. Run Algorithm 1 for $K = \frac{2\log A}{(1-\gamma)^2\varepsilon^2}$ iterations, with a learning rate $\eta = (1-\gamma)\sqrt{2\log A/K}$, and $\tau_{\text{E}} = \mathcal{O}\left(\frac{d}{(1-\gamma)^2\varepsilon^2}\log\left(\frac{B_\theta B_\varphi A}{(1-\gamma)\varepsilon}\right)\right)$ samples collected by any expert policy $\pi_{\text{E}}$. Then, the output satisfies $\mathbb{E}\left[\rho^{\pi_{\text{E}}} - \rho^{\pi^{\text{out}}}\right] = \mathcal{O}(\varepsilon)$.*

*Proof.* By Proposition 1, we have

$$\mathbb{E}\left[\rho^{\pi_{\text{E}}} - \rho^{\pi^{\text{out}}}\right] \leq \frac{1}{K}\sum_{k=1}^K \mathbb{E}[\mathcal{L}(\pi_k;Q_k)] + \frac{2}{K}\sum_{k=1}^K \mathbb{E}[\Delta(\pi_k)].$$

Using Lemma 2 with a learning rate of $\eta = (1-\gamma)\sqrt{\frac{2\log A}{K}}$ and dividing by $K$, we obtain that

$$\frac{1}{K}\sum_{k=1}^K \mathcal{L}(\pi_k;Q_k) \leq \sqrt{\frac{2\log A}{(1-\gamma)^2K}}.$$

Therefore, setting $K = \frac{2\log A}{(1-\gamma)^2\varepsilon^2}$ guarantees $\frac{1}{K}\sum_{k=1}^K \mathcal{L}(\pi_k;Q_k) \leq \varepsilon$. Then, using the high-probability bound in Lemma 3 and the fact that $\frac{1}{K}\sum_{k=1}^K \Delta(\pi_k)$ is a random variable bounded by $2(1-\gamma)^{-1}$ almost surely, we obtain the following expectation bound which holds for all $\delta > 0$,

$$\frac{1}{K}\sum_{k=1}^K \mathbb{E}[\Delta(\pi_k)] \leq \frac{1}{K} + C\sqrt{\frac{d}{(1-\gamma)^2\tau_{\text{E}}}\log\left(\frac{B_\theta B_\varphi A}{(1-\gamma)\delta\varepsilon}\right)} + \frac{2\delta}{1-\gamma},$$

for some $C \in \mathbb{R}$. Note that the choice of parameters ensures $\frac{1}{K} \leq \frac{\varepsilon}{2}$. Setting $\delta = \frac{\varepsilon(1-\gamma)}{4}$ and

$$\tau_{\text{E}} \geq \frac{2C^2 d}{(1-\gamma)^2\varepsilon^2}\log\left(\frac{B_\theta B_\varphi A}{(1-\gamma)\varepsilon}\right)$$

this bound implies that $\frac{2}{K}\sum_{k=1}^K \mathbb{E}[\Delta(\pi_k)] \leq 4\varepsilon$. Thus, we conclude that $\mathbb{E}\left[\rho^{\pi_{\text{E}}} - \rho^{\pi^{\text{out}}}\right] \leq 5\varepsilon$. $\quad\square$

## B.6 Proof of Lemma 4 (concentration general case)

Before presenting the proof of Theorem 2, we provide a bound on the covering number of the class $\mathcal{Q} \times \Pi_{\mathcal{Q}}$, where $\Pi_{\mathcal{Q}}$ is defined in Equation (4). It turns out that the covering number of this class is exponential in $K$. In the linear case, the exponential dependence in $K$ was avoided because the state-action value class is closed under addition.

**Lemma 7** (Covering number of $\Pi_{\mathcal{Q}}$)**.** *For $\epsilon > 0$, it holds that the $\epsilon$-covering number of the policy class $\Pi_{\mathcal{Q}}$ can be bounded as*

$$\mathcal{N}_\epsilon(\Pi_{\mathcal{Q}}, \|\cdot\|_1) \leq \mathcal{N}_{\frac{\epsilon}{K\eta A}}(\mathcal{Q}, \|\cdot\|_\infty)^K,$$

*where, with a slight abuse of notation, $\|\cdot\|_1$ denotes the distance defined for any $\pi, \pi' \in \Pi_{\mathcal{Q}}$ as $\|\pi - \pi'\|_1 = \sup_{x \in \mathcal{X}} \|\pi(\cdot \mid x) - \pi'(\cdot \mid x)\|_1$. Moreover,*

$$\mathcal{N}_\epsilon\Big(\mathcal{Q} \times \Pi_{\mathcal{Q}}, \|\cdot\|_{\infty,1}\Big) \leq \mathcal{N}_{\frac{\epsilon}{2K\eta A}}(\mathcal{Q}, \|\cdot\|_\infty)^{K+1}.$$

*Proof.* Let us consider two policies $\pi$ and $\pi'$ in the class $\Pi_{\mathcal{Q}}$. There exist $Q_1, \ldots, Q_K \in \mathcal{Q}$ and $Q'_1, \ldots, Q'_K \in \mathcal{Q}$ such that for any state-action pair $(x, a) \in \mathcal{X} \times \mathcal{A}$, $\pi$ and $\pi'$ can be written as

$$\pi(a \mid x) = \frac{\exp\Big(\eta \sum_{k=1}^K Q_k(x, a)\Big)}{\sum_{b \in \mathcal{A}} \exp\Big(\eta \sum_{k=1}^K Q_k(x, b)\Big)},$$

and

$$\pi'(a \mid x) = \frac{\exp\Big(\eta \sum_{k=1}^K Q'_k(x, a)\Big)}{\sum_{b \in \mathcal{A}} \exp\Big(\eta \sum_{k=1}^K Q'_k(x, b)\Big)}.$$

Let $x \in \mathcal{X}$. Using $\|\cdot\|_1 \leq \sqrt{A} \|\cdot\|$ in $\mathbb{R}^A$ and by 1-Lipschitzness of the softmax function (Lemma 15), it holds that

$$
\begin{aligned}
\|\pi(\cdot \mid x) - \pi'(\cdot \mid x)\|_1 &\leq \sqrt{A} \|\pi(\cdot \mid x) - \pi'(\cdot \mid x)\| \\
&\leq \eta\sqrt{A} \left\| \sum_{k=1}^K (Q_k(x, \cdot) - Q'_k(x, \cdot)) \right\| \\
&\leq \eta\sqrt{A} \sum_{k=1}^K \|Q_k(x, \cdot) - Q'_k(x, \cdot)\| && \text{(Triangle inequality)} \\
&\leq \eta A \sum_{k=1}^K \sup_{a \in \mathcal{A}} |Q_k(x, a) - Q'_k(x, a)| && (\|\cdot\| \leq \sqrt{A} \|\cdot\|_\infty) \\
&\leq \eta A \sup_{x \in \mathcal{X}} \left\{ \sum_{k=1}^K \sup_{a \in \mathcal{A}} |Q_k(x, a) - Q'_k(x, a)| \right\} \\
&\leq \eta A \sum_{k=1}^K \|Q_k - Q'_k\|_\infty && \text{(Triangle inequality)}.
\end{aligned}
$$

In particular, this implies

$$\max_{x \in \mathcal{X}} \|\pi(\cdot \mid x) - \pi'(\cdot \mid x)\|_1 \leq \eta A \sum_{k=1}^K \|Q'_k - Q_k\|_\infty.$$

Thus, the $\epsilon$-covering number for $\Pi_{\mathcal{Q}}$, $\mathcal{N}_\epsilon(\Pi_{\mathcal{Q}}, \|\cdot\|_1)$, is upper-bounded by the $\frac{\epsilon}{K\eta A}$-covering number of the class $\mathcal{Q}$ to the power $K$, *i.e.*, $\mathcal{N}_{\frac{\epsilon}{K\eta A}}(\mathcal{Q}, \|\cdot\|_\infty)^K$. Thus,

$$\mathcal{N}_\epsilon(\Pi_{\mathcal{Q}}, \|\cdot\|_1) \leq \mathcal{N}_{\frac{\epsilon}{K\eta A}}(\mathcal{Q}, \|\cdot\|_\infty)^K.$$

The proof is concluded by noting that the covering number increases with the precision (when $\epsilon$ decreases), and therefore, we can write

$$
\begin{aligned}
\mathcal{N}_\epsilon\Big(\mathcal{Q} \times \Pi_{\mathcal{Q}}, \|\cdot\|_{\infty,1}\Big) &\leq \mathcal{N}_{\epsilon/2}(\mathcal{Q}, \|\cdot\|_\infty)\mathcal{N}_{\epsilon/2}(\Pi_{\mathcal{Q}}, \|\cdot\|_1) \\
&\leq \mathcal{N}_{\epsilon/2}(\mathcal{Q}, \|\cdot\|_\infty)\mathcal{N}_{\frac{\epsilon}{2K\eta A}}(\mathcal{Q}, \|\cdot\|_\infty)^K \\
&\leq \mathcal{N}_{\frac{\epsilon}{2K\eta A}}(\mathcal{Q}, \|\cdot\|_\infty)^{K+1}.
\end{aligned}
$$

$\square$

Finally, the following result proves the concentration of the estimators used in Algorithm 2.

**Lemma 4.** *Let $\mathcal{Q} \subset \mathbb{R}^{\mathcal{X} \times \mathcal{A}}$ denote an arbitrary class, $\{\pi_k\}_{k=1}^K$ denote the iterates produced by Algorithm 2, and let $\Delta(\pi_k)$ be defined as in Proposition 1. Then, with probability at least $1 - \delta$, it holds that for all $k \in [K]$*

$$\Delta(\pi_k) \leq \frac{1}{K} + \sqrt{\frac{8(K+1)\log\left(2\mathcal{N}_{\frac{1-\gamma}{8K^2\eta A}}(\mathcal{Q}, \|\cdot\|_\infty)/\delta\right)}{(1-\gamma)^2 \tau_\mathrm{E}}} \, .$$

*Proof.* Note that by construction, the policy sequence $\{\pi_k\}_{k\in[K]}$ generated by Algorithm 2 belongs to the policy class $\Pi_\mathcal{Q}$. Therefore, invoking Lemma 5, we have that with probability at least $1 - \delta$, for any $k \in [K]$, it holds that

$$\Delta(\pi_k) \leq \inf_{\epsilon:\epsilon>0}\left\{\frac{4\epsilon}{1-\gamma} + \sqrt{\frac{8\log\left(2\mathcal{N}_\epsilon\left(\mathcal{Q} \times \Pi_\mathcal{Q}, \|\cdot\|_{\infty,1}\right)/\delta\right)}{(1-\gamma)^2 \tau_\mathrm{E}}}\right\} \, .$$

Therefore, choosing $\epsilon = \frac{1-\gamma}{4K}$, we get

$$\Delta(\pi_k) \leq \frac{1}{K} + \sqrt{\frac{8\log\left(2\mathcal{N}_{(1-\gamma)/4K}\left(\mathcal{Q} \times \Pi_\mathcal{Q}, \|\cdot\|_{\infty,1}\right)/\delta\right)}{(1-\gamma)^2 \tau_\mathrm{E}}}$$

$$\leq \frac{1}{K} + \sqrt{\frac{8(K+1)\log\left(2\mathcal{N}_{\frac{1-\gamma}{8K^2\eta A}}(\mathcal{Q}, \|\cdot\|_\infty)/\delta\right)}{(1-\gamma)^2 \tau_\mathrm{E}}} \, ,$$

where the last inequality follows from Lemma 7. $\qquad\square$

### B.7 Proof of Theorem 2 (sample complexity guarantee for $Q^\pi$-realizable MDPs)

We are now ready for the proof of Theorem 2, which we restate for convenience.

**Theorem 2.** *Let Assumption 2 hold. Run Algorithm 2 for $K = \frac{2\log A}{(1-\gamma)^2\varepsilon^2}$ iterations, with a learning rate $\eta = (1-\gamma)\sqrt{2\log A/K}$ and $\tau_\mathrm{E} = \mathcal{O}\left(\frac{\log A}{(1-\gamma)^4\varepsilon^4}\log\left(\frac{\mathcal{N}_{\varepsilon'}(\mathcal{Q}, \|\cdot\|_\infty)}{\varepsilon(1-\gamma)}\right)\right)$ samples collected by any expert $\pi_\mathrm{E}$, where $\varepsilon' = \left(8\sqrt{2}K^{3/2}A\log A\right)^{-1}$. Then, the output satisfies $\mathbb{E}\left[\rho^{\pi_\mathrm{E}} - \rho^{\pi^\mathrm{out}}\right] = \mathcal{O}(\varepsilon)$.*

*Proof.* Recall that by Proposition 1, we have

$$\mathbb{E}\left[\rho^{\pi_\mathrm{E}} - \rho^{\pi^\mathrm{out}}\right] \leq \frac{1}{K}\sum_{k=1}^K \mathbb{E}[\mathcal{L}(\pi_k; Q_k)] + \frac{2}{K}\sum_{k=1}^K \mathbb{E}[\Delta(\pi_k)] \, .$$

Then, by Lemma 2, it holds that

$$\frac{1}{K}\sum_{k=1}^K \mathbb{E}[\mathcal{L}(\pi_k; Q_k)] \leq \frac{\log(A)}{\eta K} + \frac{\eta}{(1-\gamma)^2} \, .$$

Moreover, by Lemma 4, with probability at least $1 - \delta$, it holds that

$$\sum_{k=1}^K \Delta(\pi_k) \leq 1 + K\sqrt{\frac{8(K+1)\log\left(2\mathcal{N}_{\frac{1-\gamma}{8K^2\eta A}}(\mathcal{Q}, \|\cdot\|_\infty)/\delta\right)}{(1-\gamma)^2 \tau_\mathrm{E}}} \, .$$

Since $\frac{1}{K}\sum_{k=1}^K \Delta(\pi_k)$ is bounded almost surely by $2(1-\gamma)^{-1}$, we have that for any $\delta > 0$

$$\frac{1}{K}\sum_{k=1}^K \mathbb{E}[\Delta(\pi_k)] \leq \frac{1}{K} + \sqrt{\frac{8(K+1)\log\left(2\mathcal{N}_{\frac{1-\gamma}{8K^2\eta A}}(\mathcal{Q}, \|\cdot\|_\infty)/\delta\right)}{(1-\gamma)^2 \tau_\mathrm{E}}} + \frac{2\delta}{1-\gamma} \, .$$

Setting $\eta = (1-\gamma)\sqrt{2\log(A)/K}$, we get

$$\mathbb{E}\Big[\rho^{\pi_{\mathrm{E}}} - \rho^{\pi^{\mathrm{out}}}\Big] \leq \sqrt{\frac{2\log(A)}{(1-\gamma)^2 K}} + \frac{2}{K} + 2\sqrt{\frac{8(K+1)\log(2\mathcal{N}_{\varepsilon'}(\mathcal{Q}, \|\cdot\|_\infty)/\delta)}{(1-\gamma)^2 \tau_{\mathrm{E}}}} + \frac{4\delta}{1-\gamma},$$

where we denoted $\varepsilon' = \frac{1}{8\sqrt{2}K^{3/2}\sqrt{\log(A)}A}$. Setting $\delta = \frac{(1-\gamma)\varepsilon}{4}$ and $K = \frac{2\log A}{(1-\gamma)^2\varepsilon^2}$, and noting that $\frac{1}{K} \leq \frac{\varepsilon}{2}$ (for $\varepsilon < 1$, $\gamma \in [0,1]$ and $A \geq 2$), we further have

$$\mathbb{E}\Big[\rho^{\pi_{\mathrm{E}}} - \rho^{\pi^{\mathrm{out}}}\Big] \leq \varepsilon + \varepsilon + C\sqrt{\frac{\log(A)}{(1-\gamma)^4\varepsilon^2\tau_{\mathrm{E}}}\log\Big(\frac{\mathcal{N}_{\varepsilon'}(\mathcal{Q}, \|\cdot\|_\infty)}{(1-\gamma)\varepsilon}\Big)} + \varepsilon,$$

for some constant $C > 0$. Finally, setting

$$\tau_{\mathrm{E}} \geq \frac{C^2\log(A)}{(1-\gamma)^4\varepsilon^4}\log\Big(\frac{\mathcal{N}_{\varepsilon'}(\mathcal{Q}, \|\cdot\|_\infty)}{(1-\gamma)\varepsilon}\Big),$$

where $\varepsilon' = \frac{(1-\gamma)^3\varepsilon^3}{32(\log A)^2 A}$ after plugging the value of $K$, we guarantee that

$$\mathbb{E}\Big[\rho^{\pi_{\mathrm{E}}} - \rho^{\pi^{\mathrm{out}}}\Big] = \mathcal{O}(\varepsilon).$$

$\square$

## B.8 Improvement for convex $\mathcal{Q}$ classes

In this section, we show that, when the class of state-action value functions $\mathcal{Q}$ is convex, we can improve the sample complexity from Theorem 2 to be of the same order as in the linear case, *i.e.*, $\mathcal{O}(\varepsilon^{-2})$ instead of $\mathcal{O}(\varepsilon^{-4})$.

**Assumption 3** (Convexity of $\mathcal{Q}$). *The class of state-action value functions $\mathcal{Q}$ is convex.*

The key observation is that, when $\mathcal{Q}$ is convex, the covering number of the induced policy class $\Pi_{\mathcal{Q}}$ can be bounded without an exponential dependence in $K$, as we show in the following result.

**Lemma 8** (Covering number of $\Pi_{\mathcal{Q}}$). *Let Assumption 3 hold. Then, for $\epsilon > 0$, the $\epsilon$-covering number of the policy class $\Pi_{\mathcal{Q}}$ can be bounded as*

$$\mathcal{N}_\epsilon(\Pi_{\mathcal{Q}}, \|\cdot\|_1) \leq \mathcal{N}_{\frac{\epsilon}{K\eta A}}(\mathcal{Q}, \|\cdot\|_\infty)$$

*where, with a slight abuse of notation, we denoted $\|\cdot\|_1$ the distance defined for any $\pi, \pi' \in \Pi_{\mathcal{Q}}$ as $\|\pi - \pi'\|_1 = \sup_{x \in \mathcal{X}} \|\pi(\cdot \mid x) - \pi'(\cdot \mid x)\|_1$. Moreover,*

$$\mathcal{N}_\epsilon\Big(\mathcal{Q} \times \Pi_{\mathcal{Q}}, \|\cdot\|_{\infty,1}\Big) \leq \mathcal{N}_{\frac{\epsilon}{K\eta A}}(\mathcal{Q}, \|\cdot\|_\infty)^2.$$

*Proof.* Let us consider two policies $\pi$ and $\pi'$ in the class $\Pi_{\mathcal{Q}}$. There exist $Q_1, \ldots, Q_K \in \mathcal{Q}$ and $Q'_1, \ldots, Q'_K \in \mathcal{Q}$ such that for any state-action pair $(x, a) \in \mathcal{X} \times \mathcal{A}$, $\pi$ and $\pi'$ can be written as

$$\pi(a \mid x) = \frac{\exp\Big(\eta \sum_{k=1}^K Q_k(x, a)\Big)}{\sum_{b \in \mathcal{A}} \exp\Big(\eta \sum_{k=1}^K Q_k(x, b)\Big)},$$

and

$$\pi'(a \mid x) = \frac{\exp\Big(\eta \sum_{k=1}^K Q'_k(x, a)\Big)}{\sum_{b \in \mathcal{A}} \exp\Big(\eta \sum_{k=1}^K Q'_k(x, b)\Big)}.$$

Let $x \in \mathcal{X}$. Using $\|\cdot\|_1 \leq \sqrt{A}\|\cdot\|$ in $\mathbb{R}^A$ and by 1-Lipschitzness of the softmax function (Lemma 15), it holds that

$$\begin{aligned}
\|\pi(\cdot \mid x) - \pi'(\cdot \mid x)\|_1 &\leq \sqrt{A}\|\pi(\cdot \mid x) - \pi'(\cdot \mid x)\| \\
&\leq \eta\sqrt{A}\left\|\sum_{k=1}^K (Q_k(x, \cdot) - Q'_k(x, \cdot))\right\| \\
&\leq \eta\sqrt{A}K\left\|K^{-1}\sum_{k=1}^K Q_k(x, \cdot) - K^{-1}\sum_{k=1}^K Q'_k(x, \cdot)\right\|.
\end{aligned}$$

At this point, we can define $\bar{Q}(x, a) = K^{-1} \sum_{k=1}^{K} Q_k(x, a)$ and $\bar{Q}'(x, a) = K^{-1} \sum_{k=1}^{K} Q'_k(x, a)$ for all $x, a \in \mathcal{X} \times \mathcal{A}$ and obtain

$$
\begin{aligned}
\|\pi(\cdot \mid x) - \pi'(\cdot \mid x)\|_1 &\leq \eta \sqrt{A} K \left\| \bar{Q}(x, \cdot) - \bar{Q}'(x, \cdot) \right\| \\
&\leq \eta A K \left\| \bar{Q}(x, \cdot) - \bar{Q}'(x, \cdot) \right\| \qquad (\|\cdot\| \leq \sqrt{A} \|\cdot\|_\infty) \\
&\leq \eta A K \left\| \bar{Q} - \bar{Q}' \right\|_\infty .
\end{aligned}
$$

At this point, notice that by convexity of $\mathcal{Q}$ we have that $\bar{Q}, \bar{Q}' \in \mathcal{Q}$. Thus , we have, the $\epsilon$-covering number for $\Pi_\mathcal{Q}$, $\mathcal{N}_\epsilon(\Pi_\mathcal{Q}, \|\cdot\|_1)$, is upper-bounded by the $\frac{\epsilon}{K\eta A}$-covering number of the class $\mathcal{Q}$, *i.e.*, $\mathcal{N}_{\frac{\epsilon}{K\eta A}}(\mathcal{Q}, \|\cdot\|_\infty)$. Thus,

$$
\mathcal{N}_\epsilon(\Pi_\mathcal{Q}, \|\cdot\|_1) \leq \mathcal{N}_{\frac{\epsilon}{K\eta A}}(\mathcal{Q}, \|\cdot\|_\infty) .
$$

The proof is concluded by noting that the covering number increases with the precision (when $\epsilon$ decreases), and therefore, we can write

$$
\begin{aligned}
\mathcal{N}_\epsilon \left( \mathcal{Q} \times \Pi_\mathcal{Q}, \|\cdot\|_{\infty, 1} \right) &\leq \mathcal{N}_{\epsilon/2}(\mathcal{Q}, \|\cdot\|_\infty) \mathcal{N}_{\epsilon/2}(\Pi_\mathcal{Q}, \|\cdot\|_1) \\
&\leq \mathcal{N}_{\epsilon/2}(\mathcal{Q}, \|\cdot\|_\infty) \mathcal{N}_{\frac{\epsilon}{K\eta A}}(\mathcal{Q}, \|\cdot\|_\infty) \\
&\leq \mathcal{N}_{\frac{\epsilon}{K\eta A}}(\mathcal{Q}, \|\cdot\|_\infty)^2 .
\end{aligned}
$$

$\square$

Importantly, the covering number of $\Pi_\mathcal{Q}$ is no longer exponential in $K$ if the class $\mathcal{Q}$ is convex. Therefore, plugging Lemma 8 into the general concentration argument in Lemma 5, we obtain the following result.

**Lemma 9.** *Let Assumption 3 hold, let $\{\pi_k\}_{k \in [K]}$ be the sequence of policies generated by Algorithm 2, and $\Delta(\pi_k)$ be defined as in Proposition 1. Then, with probability at least $1 - \delta$, for any $k \in [K]$, it holds that*

$$
\Delta(\pi_k) \leq \frac{1}{K} + \sqrt{\frac{16 \log \left( 2 \mathcal{N}_{\frac{1-\gamma}{4K^2 \eta A}}(\mathcal{Q}, \|\cdot\|_\infty)/\delta \right)}{(1-\gamma)^2 \tau_E}} .
$$

*Proof.* Invoking Lemma 5, we have that with probability at least $1 - \delta$, for any $k \in [K]$, it holds that

$$
\Delta(\pi_k) \leq \inf_{\epsilon : \epsilon > 0} \left\{ \frac{4\epsilon}{1-\gamma} + \sqrt{\frac{8 \log \left( 2 \mathcal{N}_\epsilon \left( \mathcal{Q} \times \Pi_\mathcal{Q}, \|\cdot\|_{\infty, 1} \right)/\delta \right)}{(1-\gamma)^2 \tau_E}} \right\} .
$$

Then, choosing $\epsilon = \frac{1-\gamma}{4K}$, we get

$$
\begin{aligned}
\Delta(\pi_k) &\leq \frac{1}{K} + \sqrt{\frac{8 \log \left( 2 \mathcal{N}_{(1-\gamma)/4K} \left( \mathcal{Q} \times \Pi_\mathcal{Q}, \|\cdot\|_{\infty, 1} \right)/\delta \right)}{(1-\gamma)^2 \tau_E}} \\
&\leq \frac{1}{K} + \sqrt{\frac{16 \log \left( 2 \mathcal{N}_{\frac{1-\gamma}{4K^2 \eta A}}(\mathcal{Q}, \|\cdot\|_\infty)/\delta \right)}{(1-\gamma)^2 \tau_E}} .
\end{aligned}
$$

$\square$

Finally, putting all together we can derive the following sample complexity bound for the convex case.

**Theorem 3.** *Let Assumption 3 hold, and let $\pi^{\text{out}}$ be the policy obtained running Algorithm 2 for $K = \frac{2 \log A}{(1-\gamma)^2 \varepsilon^2}$ iterations, with a learning rate $\eta = (1-\gamma) \sqrt{2 \log(A)/K}$ and*

$$
\tau_E = \mathcal{O} \left( \frac{1}{(1-\gamma)^2 \varepsilon^2} \log \left( \frac{\mathcal{N}_{\varepsilon'}(\mathcal{Q}, \|\cdot\|_\infty)}{\varepsilon(1-\gamma)} \right) \right)
$$

*samples collected by any expert* $\pi_{\mathrm{E}}$, *where* $\varepsilon' = \frac{(1-\gamma)^3 \varepsilon^3}{32(\log A)^2 A}$. *Then, the output satisfies* $\mathbb{E}\left[\rho^{\pi_{\mathrm{E}}} - \rho^{\pi^{\mathrm{out}}}\right] = \mathcal{O}(\varepsilon)$.

*Proof.* Following the arguments used in the general case, and setting $\eta = (1-\gamma)\sqrt{2\log(A)/K}$, we get

$$\mathbb{E}\left[\rho^{\pi_{\mathrm{E}}} - \rho^{\pi^{\mathrm{out}}}\right] \leq \sqrt{\frac{2\log(A)}{(1-\gamma)^2 K}} + \frac{2}{K} + 2\sqrt{\frac{16\log(2\mathcal{N}_{\varepsilon'}(\mathcal{Q}, \|\cdot\|_\infty)/\delta)}{(1-\gamma)^2 \tau_{\mathrm{E}}}} + \frac{4\delta}{1-\gamma} \,,$$

where we denoted $\varepsilon' = \frac{1}{8\sqrt{2}K^{3/2}\sqrt{\log(A)A}}$. Then, setting $\delta = \frac{(1-\gamma)\varepsilon}{4}$ and $K = \frac{2\log A}{(1-\gamma)^2\varepsilon^2}$, we further have

$$\mathbb{E}\left[\rho^{\pi_{\mathrm{E}}} - \rho^{\pi^{\mathrm{out}}}\right] \leq \varepsilon + \varepsilon + 2\sqrt{\frac{16\log(8\mathcal{N}_{\varepsilon'}(\mathcal{Q}, \|\cdot\|_\infty)/((1-\gamma)\varepsilon))}{(1-\gamma)^2 \tau_{\mathrm{E}}}} + \varepsilon.$$

Finally, setting

$$\tau_{\mathrm{E}} \geq \frac{64}{(1-\gamma)^2 \varepsilon^2} \log\left(\frac{8\mathcal{N}_{\varepsilon'}(\mathcal{Q}, \|\cdot\|_\infty)}{\varepsilon(1-\gamma)}\right),$$

we guarantee that

$$\mathbb{E}\left[\rho^{\pi_{\mathrm{E}}} - \rho^{\pi^{\mathrm{out}}}\right] = \mathcal{O}(\varepsilon).$$

$\square$

This result also provides a proof for a different sample complexity guarantee in the general case, as we show below.

**Corollary 1** (Convex-hull reduction). *Let Assumption 2 hold and let* $\pi^{\mathrm{out}}$ *be the policy obtained running Algorithm 2 for* $K = \frac{2\log A}{(1-\gamma)^2\varepsilon^2}$ *iterations, with a learning rate* $\eta = (1-\gamma)\sqrt{2\log(A)/K}$ *and*

$$\tau_{\mathrm{E}} = \mathcal{O}\left(\frac{1}{(1-\gamma)^2\varepsilon^2}\log\left(\frac{\mathcal{N}_{\varepsilon'}(\mathrm{conv}(\mathcal{Q}), \|\cdot\|_\infty)}{\varepsilon(1-\gamma)}\right)\right)$$

*samples collected by any expert* $\pi_{\mathrm{E}}$, *where* $\varepsilon' = \frac{(1-\gamma)^3\varepsilon^3}{32(\log A)^2 A}$ *and* $\mathrm{conv}(\cdot)$ *refers to taking the convex hull. Then, the output satisfies* $\mathbb{E}\left[\rho^{\pi_{\mathrm{E}}} - \rho^{\pi^{\mathrm{out}}}\right] = \mathcal{O}(\varepsilon)$.

**Remark 1.** *We note that, in general, there is no way to upper bound the covering number of* $\mathrm{conv}(\mathcal{Q})$ *in terms of that of* $\mathcal{Q}$; *the former can be much larger than the latter. Therefore, the sample complexity in Corollary 1 can be strictly worse than that in Theorem 2, depending on the structure of* $\mathcal{Q}$.

*Proof.* We follow the same steps as the proof of Theorem 3; the only difference is that we replace the convexity assumption on $\mathcal{Q}$ by working with its convex hull in the covering-number bounds.

Fix any policy $\pi$ and recall that $\widehat{\mathcal{L}}(\pi; Q)$ is linear in $Q$. Since a linear functional achieves the same supremum over a set and over its convex hull, we have

$$\sup_{Q \in \mathrm{conv}(\mathcal{Q})} \widehat{\mathcal{L}}(\pi; Q) = \sup_{Q \in \mathcal{Q}} \widehat{\mathcal{L}}(\pi; Q).$$

(Note the same argument holds for the population loss $\mathcal{L}(\pi; Q)$.) In particular, the critic update in Algorithm 2 (which selects $Q_k \in \arg\max_{Q \in \mathcal{Q}} \widehat{\mathcal{L}}(\pi_k; Q)$) is consistent with optimizing over $\mathrm{conv}(\mathcal{Q})$: it already chooses an element of $\mathrm{conv}(\mathcal{Q})$ (since $\mathcal{Q} \subseteq \mathrm{conv}(\mathcal{Q})$) achieving the same maximum value.

We check that the boundedness is preserved after taking the convex hull. By Assumption 2, we have $\|Q\|_\infty \leq (1-\gamma)^{-1}$ for all $Q \in \mathcal{Q}$. Hence for any $\widetilde{Q} \in \mathrm{conv}(\mathcal{Q})$ written as $\widetilde{Q} = \sum_i w_i Q^{(i)}$ with some discrete probability distribution $w$ and $Q^{(i)} \in \mathcal{Q}$, we have $\left\|\widetilde{Q}\right\|_\infty \leq \sum_i w_i \left\|Q^{(i)}\right\|_\infty \leq \frac{1}{1-\gamma}$. Thus the boundedness condition used in the concentration arguments continues to hold when working with $\mathrm{conv}(\mathcal{Q})$.

Next, we show that the covering-number bound of Lemma 8 continues to hold with $\mathcal{Q}$ replaced by $\mathrm{conv}(\mathcal{Q})$ on the right-hand side, even if $\mathcal{Q}$ itself is not convex. That is, for all $\epsilon > 0$,

$$\mathcal{N}_\epsilon(\Pi_\mathcal{Q}, \|\cdot\|_1) \leq \mathcal{N}_{\epsilon/(K\eta A)}(\mathrm{conv}(\mathcal{Q}), \|\cdot\|_\infty). \tag{5}$$

Indeed, take any $\pi, \pi' \in \Pi_\mathcal{Q}$. By definition, there exist $Q_1, \ldots, Q_K \in \mathcal{Q}$ and $Q'_1, \ldots, Q'_K \in \mathcal{Q}$ such that, for all $(x, a) \in \mathcal{X} \times \mathcal{A}$,

$$\pi(a \mid x) = \frac{\exp\left(\eta \sum_{k=1}^K Q_k(x, a)\right)}{\sum_{b\in\mathcal{A}} \exp\left(\eta \sum_{k=1}^K Q_k(x, b)\right)}, \qquad \pi'(a \mid x) = \frac{\exp\left(\eta \sum_{k=1}^K Q'_k(x, a)\right)}{\sum_{b\in\mathcal{A}} \exp\left(\eta \sum_{k=1}^K Q'_k(x, b)\right)}.$$

Repeating the Lipschitz argument in Lemma 8, we obtain for every $x \in \mathcal{X}$,

$$\|\pi(\cdot \mid x) - \pi'(\cdot \mid x)\|_1 \leq \eta A K \left\|\bar{Q} - \bar{Q}'\right\|_\infty,$$

where $\bar{Q} = K^{-1} \sum_{k=1}^K Q_k$ and $\bar{Q}' = K^{-1} \sum_{k=1}^K Q'_k$. Crucially, even if $\mathcal{Q}$ is not convex, we have $\bar{Q}, \bar{Q}' \in \mathrm{conv}(\mathcal{Q})$. This proves (5) exactly as in Lemma 8. Moreover, since $\mathcal{Q} \subseteq \mathrm{conv}(\mathcal{Q})$, we also have $\mathcal{N}_\epsilon(\mathcal{Q}, \|\cdot\|_\infty) \leq \mathcal{N}_\epsilon(\mathrm{conv}(\mathcal{Q}), \|\cdot\|_\infty)$. Combining with (5), we obtain for all $\epsilon > 0$,

$$\mathcal{N}_\epsilon\left(\mathcal{Q} \times \Pi_\mathcal{Q}, \|\cdot\|_{\infty,1}\right) \leq \mathcal{N}_{\epsilon/(K\eta A)}(\mathrm{conv}(\mathcal{Q}), \|\cdot\|_\infty)^2.$$

The rest of the proof is unchanged, and yields the stated condition on $\tau_\mathrm{E}$ with $\mathcal{N}_{\varepsilon'}(\mathrm{conv}(\mathcal{Q}), \|\cdot\|_\infty)$ in place of $\mathcal{N}_{\varepsilon'}(\mathcal{Q}, \|\cdot\|_\infty)$. $\qquad\square$

## B.9 Different sample complexity guarantee for finite $\mathcal{Q}$ classes

In this section, we provide an alternative sample complexity guarantee for Algorithm 2 in the special case where the value-function class $\mathcal{Q}$ is finite. The key observation is that when $\mathcal{Q}$ is finite, the set of policies that can be produced by Algorithm 2 up to iteration $K$ is also finite and can be controlled by a simple counting argument. This allows us to avoid the covering-number bound of Lemma 4 and obtain a dependence in $\tau_\mathrm{E}$ of order $\tilde{\mathcal{O}}(\varepsilon^{-2})$ (up to logarithmic factors), albeit at the cost of a worst dependency in the size of the class $\mathcal{Q}$, which we discuss later. We start with two standard combinatorial lemmas, that we prove here for completeness.

**Lemma 10** (Stars and bars). *Let $m \geq 1$ and $t \geq 0$ be integers. The number of integer-valued vectors $n = (n_1, \ldots, n_m) \in \mathbb{N}^m$ such that $\sum_{j=1}^m n_j = t$ is*

$$\left|\left\{n \in \mathbb{N}^m : \textstyle\sum_{j=1}^m n_j = t\right\}\right| = \binom{t + m - 1}{m - 1}.$$

*Proof.* Consider the set $\mathcal{S}_{t,m}$ of strings of length $t + m - 1$ over the alphabet $\{\star, |\}$ containing exactly $t$ symbols $\star$ and exactly $m - 1$ symbols $|$. Clearly, $|\mathcal{S}_{t,m}| = \binom{t+m-1}{m-1}$ since specifying such a string is equivalent to choosing the $(m - 1)$ positions of the bars among $t + m - 1$ positions.

We construct a bijection between $\mathcal{S}_{t,m}$ and the set $\mathcal{N}_{t,m} = \{n \in \mathbb{N}^m : \sum_{j=1}^m n_j = t\}$. Given a string $s \in \mathcal{S}_{t,m}$, read it from left to right and let $n_j(s)$ be the number of $\star$ symbols occurring between the $(j - 1)$-th bar and the $j$-th bar, with the convention that the 0-th bar is placed before the first character and the $m$-th bar is placed after the last character. This produces a vector

$$n(s) = (n_1(s), \ldots, n_m(s)) \in \mathbb{N}^m.$$

By construction, the total number of stars in the string is $t$, hence $\sum_{j=1}^m n_j(s) = t$, so $n(s) \in \mathcal{N}_{t,m}$.

Conversely, given any $n = (n_1, \ldots, n_m) \in \mathcal{N}_{t,m}$, define a string $s(n) \in \mathcal{S}_{t,m}$ by concatenating $n_1$ stars, then a bar, then $n_2$ stars, then a bar, and so on, ending with $n_m$ stars:

$$s(n) = \underbrace{\star\cdots\star}_{n_1} | \underbrace{\star\cdots\star}_{n_2} | \cdots | \underbrace{\star\cdots\star}_{n_m}.$$

This string has exactly $t = \sum_{j=1}^m n_j$ stars and $m - 1$ bars, so $s(n) \in \mathcal{S}_{t,m}$. Finally, it is clear from the constructions that $n(s(n)) = n$ for all $n \in \mathcal{N}_{t,m}$ and that $s(n(s)) = s$ for all $s \in \mathcal{S}_{t,m}$. Therefore $s \mapsto n(s)$ is a bijection between $\mathcal{S}_{t,m}$ and $\mathcal{N}_{t,m}$, and we conclude

$$|\mathcal{N}_{t,m}| = |\mathcal{S}_{t,m}| = \binom{t + m - 1}{m - 1}.$$

$\qquad\square$

**Lemma 11** (Hockey-stick identity). *Let $m \geq 1$ and $K \geq 0$ be integers. Then*

$$\sum_{t=0}^{K} \binom{t+m-1}{m-1} = \binom{K+m}{m}.$$

*Proof.* We give a short proof by induction on $K$ using Pascal's identity. For $K = 0$, the left-hand side equals $\binom{m-1}{m-1} = 1$ and the right-hand side equals $\binom{m}{m} = 1$, so the identity holds.

Assume the identity holds for some $K \geq 0$. Then

$$\sum_{t=0}^{K+1} \binom{t+m-1}{m-1} = \sum_{t=0}^{K} \binom{t+m-1}{m-1} + \binom{K+1+m-1}{m-1}.$$

By the induction hypothesis, the first sum equals $\binom{K+m}{m}$. Hence

$$\sum_{t=0}^{K+1} \binom{t+m-1}{m-1} = \binom{K+m}{m} + \binom{K+m}{m-1}.$$

Applying Pascal's identity $\binom{N}{r} + \binom{N}{r-1} = \binom{N+1}{r}$ with $N = K + m$ and $r = m$, we obtain

$$\binom{K+m}{m} + \binom{K+m}{m-1} = \binom{K+m+1}{m} = \binom{(K+1)+m}{m}.$$

This is exactly the desired identity for $K + 1$, completing the induction. $\square$

We also provide an upper bound on a binomial coefficient that will be useful later.

**Lemma 12** (Binomial coefficient upper bounds). *Let $m \geq 1$ and $K \geq 0$ be integers. Then*

$$\binom{K+m}{m} \leq \left( \frac{e(K+m)}{m} \right)^m.$$

*In particular,*

$$\log \binom{K+m}{m} \leq m \log \left( \frac{e(K+m)}{m} \right).$$

*Proof.* We start from the factorial expression

$$\binom{K+m}{m} = \frac{(K+m)!}{K!\,m!} = \frac{\prod_{j=1}^{m}(K+j)}{m!}.$$

Since $K + j \leq K + m$ for all $j \in [m]$, we have

$$\prod_{j=1}^{m}(K+j) \leq (K+m)^m. \tag{6}$$

Next we lower bound $m!$. Using the integral bound

$$\sum_{j=1}^{m} \log j \geq \int_{1}^{m} \log x \, \mathrm{d}x = [x \log x - x]_{1}^{m} = m \log m - m + 1,$$

we obtain

$$\log(m!) \geq m \log m - m + 1$$

which implies

$$m! \geq e^{m \log m - m + 1} = e \left( \frac{m}{e} \right)^m \geq \left( \frac{m}{e} \right)^m.$$

Combining this with (6) yields

$$\binom{K+m}{m} \leq \frac{(K+m)^m}{(m/e)^m} = \left( \frac{e(K+m)}{m} \right)^m,$$

which proves the first inequality. Taking $\log$ on both sides gives the second. $\square$

We are now ready to state and prove the main result of this section.

**Theorem 4.** *Let Assumption 2 hold and assume that $\mathcal{Q}$ is finite with cardinality $m = |\mathcal{Q}| < \infty$ and $A \geq 3$. Run Algorithm 2 for $K = \frac{2 \log A}{(1-\gamma)^2 \varepsilon^2}$ iterations, with a learning rate $\eta = (1 - \gamma)\sqrt{\frac{2 \log A}{K}}$ and $\tau_{\mathrm{E}} = \mathcal{O}\left(\frac{m}{(1-\gamma)^2 \varepsilon^2} \log\left(\frac{\log A}{(1-\gamma)\varepsilon}\right)\right)$ samples collected by any expert $\pi_{\mathrm{E}}$. Then, the output satisfies $\mathbb{E}\left[\rho^{\pi_{\mathrm{E}}} - \rho^{\pi^{\mathrm{out}}}\right] = \mathcal{O}(\varepsilon)$.*

**Remark 2.** *The result above combined with Theorem 2 shows that, for finite $\mathcal{Q}$, Algorithm 2 returns an $\mathcal{O}(\varepsilon)$-optimal policy with a number of expert trajectories scaling as*

$$\tau_{\mathrm{E}} = \tilde{\mathcal{O}}\left(\min\left(\frac{m}{(1-\gamma)^2 \varepsilon^2} \log\left(\frac{\log A}{(1-\gamma)\varepsilon}\right), \frac{\log A}{(1-\gamma)^4 \varepsilon^4} \log\left(\frac{m}{(1-\gamma)\varepsilon}\right)\right)\right).$$

*This shows that Theorem 4 is meaningful only when $m$ is relatively small, because we are trading an exponentially worse dependence in $m$ for a better polynomial dependence in $\varepsilon^{-1}$.*

*Proof.* We follow the same proof structure as the other theorems. By Proposition 1, we have

$$\mathbb{E}\left[\rho^{\pi_{\mathrm{E}}} - \rho^{\pi^{\mathrm{out}}}\right] \leq \frac{1}{K}\sum_{k=1}^{K} \mathbb{E}[\mathcal{L}(\pi_k; Q_k)] + \frac{2}{K}\sum_{k=1}^{K}\mathbb{E}[\Delta(\pi_k)],$$

Then, by Lemma 2, it holds that

$$\frac{1}{K}\sum_{k=1}^{K} \mathbb{E}[\mathcal{L}(\pi_k; Q_k)] \leq \frac{\log(A)}{\eta K} + \frac{\eta}{2(1-\gamma)^2}.$$

Let $\mathcal{Q} = \{Q^{(1)}, \ldots, Q^{(m)}\}$. For any vector $n = (n_1, \ldots, n_m) \in \mathbb{N}^m$, and any state-action pair $(x, a)$, define the policy $\pi_n$ by

$$\pi_n(a \mid x) = \frac{\exp\left(\eta \sum_{j=1}^{m} n_j Q^{(j)}(x, a)\right)}{\sum_{b \in \mathcal{A}} \exp\left(\eta \sum_{j=1}^{m} n_j Q^{(j)}(x, b)\right)}.$$

Define the (finite) policy set

$$\Pi_{K,m} = \left\{\pi_n : n \in \mathbb{N}^m, \sum_{j=1}^{m} n_j \leq K\right\}.$$

Note that for any $k \in [K]$, $\pi_k \in \Pi_{K,m}$. Indeed, Algorithm 2 computes policies of the form

$$\pi_k(a \mid x) = \frac{\exp\left(\eta \sum_{i=1}^{k-1} Q_i(x, a)\right)}{\sum_{b \in \mathcal{A}} \exp\left(\eta \sum_{i=1}^{k-1} Q_i(x, b)\right)}.$$

Since each $Q_i \in \mathcal{Q}$, there exists a (random) count vector $n^{(k)} \in \mathbb{N}^m$ such that $n_j^{(k)}$ equals the number of indices $i \in \{1, \ldots, k-1\}$ with $Q_i = Q^{(j)}$. Then $\sum_{j=1}^{m} n_j^{(k)} = k - 1 \leq K$ and

$$\sum_{i=1}^{k-1} Q_i(x, a) = \sum_{j=1}^{m} n_j^{(k)} Q^{(j)}(x, a),$$

which shows $\pi_k = \pi_{n^{(k)}} \in \Pi_{K,m}$. Next, we bound $|\Pi_{K,m}|$. By Lemma 10, for a fixed integer $t \geq 0$, the number of vectors $n \in \mathbb{N}^m$ satisfying $\sum_{j=1}^{m} n_j = t$ is $\binom{t+m-1}{m-1}$. Therefore, by Lemma 11,

$$|\Pi_{K,m}| \leq \sum_{t=0}^{K} \binom{t+m-1}{m-1} = \binom{K+m}{m}.$$

Fix $\pi \in \Pi_{K,m}$ and $Q \in \mathcal{Q}$. Recall that

$$\widehat{\mathcal{L}}(\pi; Q) = \frac{1}{\tau_{\mathrm{E}}} \sum_{i=1}^{\tau_{\mathrm{E}}} \left( Q(X_{\mathrm{E}}^i, A_{\mathrm{E}}^i) - Q(X_{\mathrm{E}}^i, \pi) \right), \quad \mathcal{L}(\pi; Q) = \mathbb{E}_{(X,A) \sim \mu_{\pi_{\mathrm{E}}}} [Q(X, A) - Q(X, \pi)].$$

Define the i.i.d. random variables

$$Z_i = Q(X_{\mathrm{E}}^i, A_{\mathrm{E}}^i) - Q(X_{\mathrm{E}}^i, \pi).$$

By Assumption 2, $\|Q\|_\infty \leq \frac{1}{1-\gamma}$, hence $|Z_i| \leq \frac{2}{1-\gamma}$. By Hoeffding's inequality (Lemma 13), for any $t > 0$,

$$\mathbb{P}\left[ \left| \widehat{\mathcal{L}}(\pi; Q) - \mathcal{L}(\pi; Q) \right| > t \right] \leq 2 \exp\left( -\frac{t^2 \tau_{\mathrm{E}} (1-\gamma)^2}{8} \right).$$

Let $M = |\Pi_{K,m}| \, |\mathcal{Q}| \leq m\binom{K+m}{m}$. A union bound over all pairs $(\pi, Q) \in \Pi_{K,m} \times \mathcal{Q}$ yields that with probability at least $1 - \delta$,

$$\sup_{\pi \in \Pi_{K,m}} \sup_{Q \in \mathcal{Q}} \left| \widehat{\mathcal{L}}(\pi; Q) - \mathcal{L}(\pi; Q) \right| \leq \sqrt{\frac{8 \log(2M/\delta)}{(1-\gamma)^2 \tau_{\mathrm{E}}}}.$$

Thus, on this event, for every $\pi \in \Pi_{K,m}$ we have

$$\Delta(\pi) = \sup_{Q \in \mathcal{Q}} \left| \widehat{\mathcal{L}}(\pi; Q) - L(\pi; Q) \right| \leq \sqrt{\frac{8 \log(2M/\delta)}{(1-\gamma)^2 \tau_{\mathrm{E}}}}.$$

Since $\pi_k \in \Pi_{K,m}$ for all $k \in [K]$, we conclude that on the same event, $\Delta(\pi_k)$ is bounded by the same quantity for all $k \in [K]$. Furthermore, note that for any $\pi$, $\Delta(\pi) \leq \frac{2}{1-\gamma}$ almost surely. Therefore, for every $k$,

$$\frac{2}{K} \sum_{k=1}^{K} \mathbb{E}[\Delta(\pi_k)] \leq 2\sqrt{\frac{8 \log(2M/\delta)}{(1-\gamma)^2 \tau_{\mathrm{E}}}} + \frac{4\delta}{1-\gamma}.$$

Finally, setting $\eta = (1-\gamma)\sqrt{\frac{2 \log A}{K}}$, we get

$$\mathbb{E}\left[ \rho^{\pi_{\mathrm{E}}} - \rho^{\pi^{\mathrm{out}}} \right] \leq \sqrt{\frac{2 \log A}{(1-\gamma)^2 K}} + 2\sqrt{\frac{8 \log(2M/\delta)}{(1-\gamma)^2 \tau_{\mathrm{E}}}} + \frac{4\delta}{1-\gamma}.$$

By Lemma 12, $\log M \leq m \log(4K)$. Setting $K = \frac{2 \log A}{(1-\gamma)^2 \varepsilon^2}$ and $\delta = \frac{(1-\gamma)\varepsilon}{4}$ yields

$$\mathbb{E}\left[ \rho^{\pi_{\mathrm{E}}} - \rho^{\pi^{\mathrm{out}}} \right] \leq \varepsilon + 2\sqrt{\frac{8m}{(1-\gamma)^2 \tau_{\mathrm{E}}} \log\left( \frac{64 \log A}{(1-\gamma)^3 \varepsilon^3} \right)} + \varepsilon.$$

It remains to choose $\tau_{\mathrm{E}}$ so that the remaining term is less than $\varepsilon$, *i.e.*,

$$\tau_{\mathrm{E}} \geq \frac{Cm}{(1-\gamma)^2 \varepsilon^2} \log\left( \frac{\log A}{(1-\gamma)\varepsilon} \right),$$

for some constant $C > 0$. With this choice, we obtain $\mathbb{E}\left[ \rho^{\pi_{\mathrm{E}}} - \rho^{\pi^{\mathrm{out}}} \right] = \mathcal{O}(\varepsilon)$, concluding the proof. $\qquad\square$

# C    Technical tools

**Lemma 13** (Hoeffding's inequality; Vershynin, 2018, Theorem 2.2.6). *Let $X_1, \ldots, X_n$ be independent random variables such that $|X_i| \leq M$ for all $i$. Then, for any $t > 0$,*

$$\mathbb{P}\left[\left|\frac{1}{n}\sum_{i=1}^{n}(X_i - \mathbb{E}[X_i])\right| > t\right] \leq 2e^{-\frac{nt^2}{2M^2}}.$$

**Lemma 14** (Simplified version of Orabona, 2023, Theorem 6.10). *Let us consider a non-empty closed convex set $V$, an arbitrary sequence of adaptively chosen loss vectors $(\ell_k)_{k=1}^{K}$ such that $\|\ell_k\|_\infty \leq \ell_{\max}$, and let $D\colon V \times \text{int}(V) \to \mathbb{R}$ be a Bregman divergence induced by a $\lambda$-strongly convex function in the $\ell_1$-norm. Then, for all $u \in V$, the sequence $(x_k)_{k=1}^{K}$ generated for any $k$ as*

$$x_{k+1} = \underset{v \in V}{\arg\min}\left\{\langle \ell_k, v \rangle + \frac{1}{\eta}D(v, x_k)\right\}$$

*for an arbitrary initial $x_1$ satisfies*

$$\sum_{k=1}^{K}\langle \ell_k, x_k - u \rangle \leq \frac{D(u, x_1)}{\eta} + \frac{\eta K \ell_{\max}^2}{2\lambda}.$$

**Lemma 15** (Gao and Pavel, 2018, Proposition 4). *For any $\eta > 0$, let the softmax function be defined for any $z \in \mathbb{R}^n$ as*

$$\text{softmax}(z) = \left(\frac{e^{\eta z_i}}{\sum_{j=1}^{n} e^{\eta z_j}}\right)_{i \in [n]}.$$

*Then, the softmax function is $\eta$-Lipschitz with respect to $\|\cdot\|_2$. That is, for any $z, z' \in \mathbb{R}^n$, we have*

$$\|\text{softmax}(z) - \text{softmax}(z')\|_2 \leq \eta \|z - z'\|_2.$$

**Lemma 16** (Covering number of a Euclidean ball; Vershynin, 2018, Corollary 4.2.11). *For $\epsilon > 0$, the $\epsilon$-covering number of the Euclidean ball of radius $R$ in $\mathbb{R}^d$, $\mathfrak{B}(R)$, is bounded as*

$$\mathcal{N}_\epsilon(\mathfrak{B}(R), \|\cdot\|) \leq \left(1 + \frac{2R}{\epsilon}\right)^d.$$

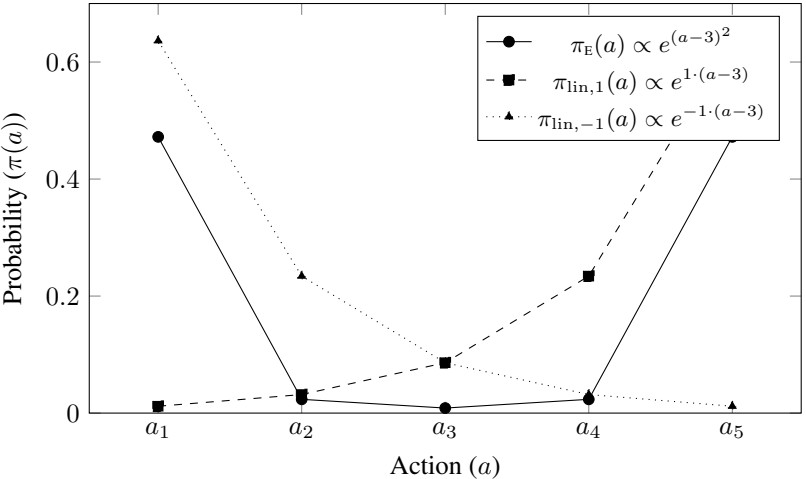

Figure 3: Comparison of linear and quadratic softmax policies with $A = 5$ actions and features $\varphi(a) = a - 3$.

## D   On the guarantees of misspecified BC in linear $Q^\pi$-realizable MDPs

It is natural to question whether existing bounds for behavioral cloning (BC) in misspecified settings (*e.g.*, Rohatgi et al., 2025; Foster et al., 2024) offer satisfactory sample complexity guarantees for imitating an arbitrarily complex expert within a linear $Q^\pi$-realizable MDP. This section presents a negative result, demonstrating that the approximation error incurred by BC, when restricted to a linear softmax policy class (denoted $\Pi_{\text{lin}}$), can be large even in a simple linear $Q^\pi$-realizable MDP.

Consider a single-state MDP defined as follows. Let $A \in \mathbb{N}^*$ be the number of actions, with the action space $\mathcal{A} = \{1, \ldots, A\}$. For each action $a \in \mathcal{A}$, there is a scalar feature $\varphi(a) = -\frac{A}{2} + a \in \mathbb{R}$. To ensure the MDP is linear $Q^\pi$-realizable, the true reward function is $r_{\text{true}}(a) = \zeta\varphi(a)$ for some parameter $\zeta \in \mathbb{R}$ unknown to the learner. We define a *softmax quadratic* expert policy $\pi_{\text{E}}$ as

$$\pi_{\text{E}}(a) = \frac{\exp\left(\varphi(a)^2\right)}{\sum_{b \in \mathcal{A}} \exp\left(\varphi(b)^2\right)} \ .$$

This expert policy assigns the highest probability to extremal actions (*i.e.*, $a = 1$ and $a = A$). In contrast, linear softmax policies $\pi \in \Pi_{\text{lin}}$ (which are commonly used for BC in feature-based settings) are inherently designed to produce monotonic probability distributions over the action space when features are ordered (*i.e.*, for actions $a, a' \in \mathcal{A}$ with $a' > a$, either $\pi(a) \leq \pi(a')$ or $\pi(a) \geq \pi(a')$). Consequently, for $A > 2$, no policy in $\Pi_{\text{lin}}$ can achieve a small Hellinger distance to this softmax quadratic expert. We illustrate this in Figure 3, where we compare the softmax quadratic expert with two linear softmax policies. Due to the monotonicity constraint, the linear softmax policies are unable to approximate the expert policy everywhere.

It remains an open question whether behavioral cloning analyses can be refined to better leverage the underlying MDP structure in such misspecified scenarios. Specifically, for the constructed example, it would be advantageous if the misspecification error in existing bounds were characterized in terms of feature expectations (*e.g.*, $\sum_{a \in \mathcal{A}} \pi(a)\varphi(a)$) rather than state-action distributions.

# E  Additional experiments

## E.1  `BC` with a simple expert class can outperform `SPOIL`

We consider a linear MDP (which is a special case of linear $Q^\pi$-realizability) with features of dimension $d = 3$. The expert class, denoted by $\Pi^E_{\text{lin,small}}$, is the class of softmax linear policies with features corresponding to the linear MDP features. That is, `BC` is given the most compact representation possible. On the other hand, the $\mathcal{Q}_{\text{lin,large}}$ class is created using the linear MDP features, plus a set of $20d$ redundant features. It follows that the complexity of the $\mathcal{Q}$ function class is larger than the expert policy function class and therefore `BC` is expected to outperform `SPOIL` on this instance. This fact is confirmed by the experiment shown in Figure 4.

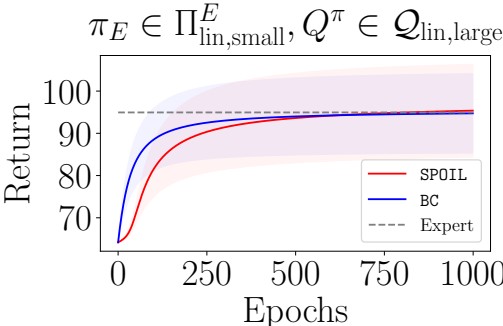

Figure 4:  Instance in which `BC` with a simple expert class can outperform `SPOIL`.

## E.2  Comparison with `IQ-Learn` in the linear case

We make a comparison with `IQ-Learn` in the Linear MDP described in Section 5. In this experiment, we consider a linear Q-function class for both SPOIL and IQ-Learn, as the environment has this structure. The results in Figure 5 show that, like `SPOIL`, `IQ-Learn`'s performance is unaffected by the complexity of the expert class. However, `SPOIL` still reaches a higher cumulative reward in this environment. To ensure this is a fair comparison, we used the same learning rate for the policy updates in `SPOIL` and `IQ-Learn`. Moreover, we tested different choices for `IQ-Learn`'s critic learning rate, and we report here the best results we could obtain.

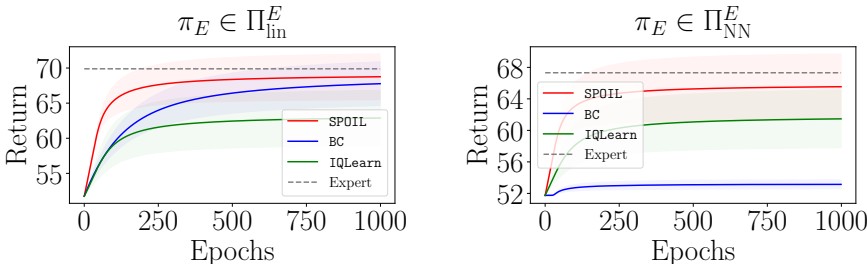

Figure 5:  Experiments in continuous-state domains. Curves are averaged across 10 seeds.

## E.3  Omitted experimental details

For the first experiment shown in Figure 1, one may wonder if the underperformance of behavioural cloning might be due to underoptimizing the empirical log-likelihood. We have ruled out this possibility by going into great lengths to optimize the likelihood, and in fact the log-likelihood has approached its minimum value of zero very closely in our experiment (meaning that the probability assigned to the actions seen in the expert dataset is almost 1). For this optimization task, we have used Adam with default parameter settings. For the experiments in Figure 2, algorithms are

implemented using a shared neural network architecture consisting of 3 layers with 64 neurons per layer. This architecture matches the one used for experiments in the same environments by Garg et al. (2021). For behavioral cloning, we employ a separate three-layer multilayer perceptron with 128 neurons per layer. Implementations of `IQ-Learn` and $P^2$`IL` utilize their original hyperparameter configurations as reported in their respective publications. All networks are optimized using the Adam optimizer (Kingma and Ba, 2014) with a learning rate of $5 \times 10^{-3}$ and default momentum parameters ($\beta_1 = 0.9, \beta_2 = 0.999$). The implementations are built using PyTorch (Paszke et al., 2019).

For algorithms with a primal-dual structure (*i.e.*, `IQ-Learn`, $P^2$`IL`, and `SPOIL`), the policy update is performed using a `Soft DQN`-style update (*c.f.* Haarnoja et al., 2017) with a fixed temperature parameter. These three algorithms thus only differ in terms of their Q-value updates, and thus this experiment serves to assess the effectiveness of the novel critic loss introduced in this work.

