# OpenReview forum: "Inverse Q-Learning Done Right: Offline Imitation Learning in $Q^\pi$-Realizable MDPs"
_NeurIPS.cc/2025/Conference — NeurIPS 2025 poster_

### Official Review · Reviewer_UNX4 · 2025-06-08

**Clarity:** 4
**Significance:** 3
**Originality:** 3
**Rating:** 5
**Confidence:** 3

**Summary:**

The RL imitation learning problem is considered where a dataset is provided based on an expert policy and the objective is to output a policy whose expected return is nearly as large. Previous works assumed that the expert policy belonged to a policy class which was given to the learner and provided bounds based on the complexity of the policy class. This work instead considers an assumption on the structure of the MDP, namely that the state-action value function of all policies belong to a function class ($Q^\pi$-realizability) given to the learner. The authors prove that their algorithm (SPOIL) requires samples from the expert policy that scales polynomially in the quantities of interest. This result is interesting because there exist MDPs that satisfy the $Q^\pi$-realizability assumption; however, a complex policy class can be defined, which implies that the SPOIL algorithm will require less samples than previous policy based algorithms. The theoretical results are supported with experiments.

**Questions:**

Please see the Weaknesses section above.

**Ethical Concerns:**

["NO or VERY MINOR ethics concerns only"]

**Final Justification:**

The authors have addressed all of my concerns. The work provides the first positive result under the $Q^\pi$-realizability assumption. The assumption is very commonly considered in other works and thus will be an interesting result to the RL theory community.

**Limitations:**

Yes

**Paper Formatting Concerns:**

No Concerns

**Quality:**

3

**Strengths And Weaknesses:**

$\textbf{Strengths:}$
1. The contribution is clearly stated and compared to previous results.
2. The work provides the first positive result under the $Q^\pi$-realizability assumption, which is a weaker structural assumption than previous works.
3. Experiments are provided supporting the theory.

$\textbf{Weaknesses:}$
1. I think the performance difference lemma (Lemma 1) should have a factor of $1/(1-\gamma)$ multiplying the right side of the equality based on the current definition of $\rho^\pi$. However, I think that $\rho^\pi$ was meant to be defined with a factor of $(1-\gamma)$ in front since that is how it is used throughout (for instance see the equation below line 148). I did not go through all the proof details in the appendix; however, if $\rho^\pi$ is indeed incorrectly defined then the authors should make sure this does not affect their final result, or update it appropriately.
2. Does Assumption 1 need realizability of $\textit{all}$ policies, or is it enough to have it for just stationary policies, which I think is more commonly assumed?
3. The norm notation $||\cdot||$ is used in Assumption 1 but I do not think it is defined anywhere. I suspect it is the 2-norm, but it should be made explicit.
4. The objective (eq 2) is very similar to that of offline policy optimization. It seems that a result from the offline policy optimization setting would imply a result in the imitation learning setting since there the policy $\pi$ that is output satisfies $|\max_{\pi'}\rho^{\pi'} - \rho^\pi| \le \epsilon$. It would be useful if a discussion could be added (at least in the appendix) about how the offline imitation learning setting relates to the offline policy optimization setting.
5. There are multiple typos that should be fixed. For example: Line 287, Line 327, Line 336
6. The conclusion states the work does not require trajectory access while some other works do. I think this is the first time trajectory access is discussed in the paper and it is challenging to understand what is meant by trajectory access and why previous works needed this since it is not mentioned in the body. It would be helpful if trajectory access was discussed in more detail in the body when comparing to previous results.

---

> ### Author Rebuttal · Authors · 2025-07-30
>
> Thank you for your encouraging review and your careful reading of our paper! Please find our answers below.
>
> 1. You are correct, there are some inconsistencies with the effective horizon factor that we have also noticed. They will be fixed in the final version.
> 2. The realizability assumption only needs to hold for the policies generated by SPOIL, not for all possible policies. In the linear setting, this amounts to requiring realizability only for the softmax linear policies. This is a significantly milder requirement than in related RL theory works, which assume linear $Q^\pi$-realizability (e.g., [1, 2]). Moreover, we do not require the realizability of the expert Q value. We will add a remark on this in the main text.
> 3. Yes, it is the Euclidean norm; we will define it explicitly.
> 4. The key difference is that in offline policy optimization, the learner observes a dataset containing the rewards of the observed state-action pairs. The rewards are not observed in offline imitation learning. We will comment on these differences in the related works section.
> 5. Thank you for spotting these! We will fix them.
> 6. This is a good point; we will explain more carefully what we mean by trajectory access.
>
> Thank you again for your valuable feedback. We are happy to discuss further if you have any other questions.
>
> **References**
>
> [1] Weisz, G., György, A., and Szepesvári, C. (2023). Online RL in Linearly $q^\pi$-Realizable MDPs Is as Easy as in Linear MDPs If You Learn What to Ignore. Advances in Neural Information Processing Systems, 36, 59172-59205.
>
> [2] Mhammedi, Z.. (2025). Sample and Oracle Efficient Reinforcement Learning for MDPs with Linearly-Realizable Value Functions. Thirty Eighth Conference on Learning Theory.

---

> > ### Comment · Reviewer_UNX4 · 2025-08-02
> >
> > You have clarified my concerns for 1. - 5.
> >
> > Regarding 6., can you please explain here what you mean by trajectory access? Also, why other works needed it and why you do not?

---

> > > ### Author Response · Authors · 2025-08-02
> > > **On the trajectory access**
> > >
> > > Dear reviewer,
> > >
> > > thanks a lot for your answer! We are happy to hear that your questions 1-5 have been addressed.
> > >
> > > When we say that an algorithm has trajectory access, we mean that such algorithm can deploy any policy in the environment $\mathcal{M}$, i.e. the one where the expert acted to collect the expert dataset.
> > >
> > > More formally, trajectory access means that for any policy $\pi$ and any horizon $H \in \mathbb{N}$, the learner can sample a sequence $(X_0, A_0, …, X_H, A_H)$ such that
> > > $X_0 \sim \nu_0$, $A_h \sim \pi(\cdot|X_h)$, $X_{h+1} \sim P(\cdot|X_h, A_h)$, for $h\in \{ 0, \dots, H\}$.
> > >
> > > Therefore, with trajectory access a learner can sample state action pairs from occupancy measures different from the expert occupancy measure.
> > >
> > > Previous works assumed trajectory access to learn with more limited expert information, such as observing a dataset containing only the states visited by the expert and not the chosen actions. It has also been shown that trajectory access is necessary to learn under these conditions.
> > >
> > > Instead, in the current submission, we can avoid the trajectory access assumption because our expert dataset contains the expert actions. Notice that this is the standard setting adopted in offline imitation learning/behavioral cloning analysis (see for example [1]).
> > >
> > > Thanks a lot for getting the discussion started. We are happy to answer any further question you might have.
> > >
> > > Best,
> > > Authors
> > >
> > > **Reference**
> > >
> > > [1] Foster et al., 2024 “Is Behavioral Cloning all you need ? Understanding Horizon in Imitation Learning"

---

> > > > ### Comment · Reviewer_UNX4 · 2025-08-04
> > > >
> > > > This has clarified things, thank you. I am happy to keep my same rating.

---

> > > > > ### Author Response · Authors · 2025-08-04
> > > > >
> > > > > Dear reviewer,
> > > > >
> > > > > we are glad that this has clarified things. Thanks for your answer and for maintaining your positive assessment.
> > > > >
> > > > > Best, Authors

---

### Official Review · Reviewer_Mw96 · 2025-06-26

**Clarity:** 2
**Significance:** 2
**Originality:** 3
**Rating:** 4
**Confidence:** 3

**Summary:**

In this paper a core challenge in imitation learning is addressed, inconsistent expert behavior. Rather than assuming the expert policy lies within a known class of optimal policies, a common but often unrealistic assumption especially in scenarios where expert behavior may be irrational, or influenced by external factors (as with human experts). This issue is known as misspecification. To address this, the authors propose a formulation that avoids step-by-step behavioral imitation and instead aims to match the expert's performance outcome. This frees the learning agent from inconsistent behaviors that may be apparent in the expert policy and focus instead on learning policies that achieve comparable returns.

The proposed algorithm, SPOIL (Saddle-Point Offline Imitation Learning), is based on an actor-critic model structured as a min-max optimization of an empirically defined loss function to that fits the previous narrative. The authors implement this approach for both linear Q-functions, which are straightforward to analyze and allow efficient closed form updates; and for general function approximators, such as neural networks, which are more flexible and realistic but require optimization and introduce additional computational complexity.
They provide experimental results, first comparing both variants of SPOIL to BC in a controlled MDP environment to showcase its training and final converging superiority, and then proceed to standard gym environments, showcasing their performance with different sizes of expert data in contrast to other high-performing approaches.

**Questions:**

1. Can you elaborate on your experimental results?

2. What are the key differences with each one of the baselines?

3. In which ways are the observed improvement of your algorithm significant?

4. Can you comment on the lack of experiments in other, less toy, domains?

Suggestions:

More challenging environments like Mujoco or Atari games would better fit your narrative. The authors could also take a look at this survey paper for environments used in imitation learning: https://arxiv.org/abs/2404.19456

A simple example could have helped clarify the core idea early in the introduction.

**Ethical Concerns:**

["NO or VERY MINOR ethics concerns only"]

**Final Justification:**

I appreciate the authors clarifications in the rebuttal. The main issue remaining in my opinion is that they have experimented with small-scale / toy experiments only. Moreover, many clarifications they provided should have been in the paper to begin with - in case of acceptance, they must be included.

**Limitations:**

yes

**Quality:**

3

**Strengths And Weaknesses:**

Strengths

1. The approach has an interesting premise, especially as the community strives to get closer to a more realistic representation of imitation learning, where there is a plethora of ways for policies to get inconsistent or unstructured in general.

2. Solid theoretical work, supported by well-structured theoretical analysis, including performance guarantees

3. The SPOIL algorithm is appropriate for the presented problem, including both a linear version that is easier to implement and observe results, as well as a more general version for modern deep learning approaches.

4. The structure and presentation of the paper is clear and overall straight forward.

Weaknesses

1.	The authors present experimental results, but there’s little explanation of what are the main differences between SPOIL and the baseline methods used.

2.	The chosen environments are all toy environments, simple by their nature and low dimensional. These are fine for initial testing but not enough to support broader claims, especially when making state-of-the-art comparisons.

3.	Since the authors compare their work to BC, it would be fitting to also discuss how SPOIL handles or is affected by covariate shift.

4.	The introduction packs in a lot of background without examples or intuitive explanations.

---

> ### Author Rebuttal · Authors · 2025-07-30
>
> Thank you for your feedback and for recognizing the solid theoretical work and clear structure of our paper. We address your questions and clarify the scope of our contributions below. We would be grateful if you might consider increasing your score in case you are satisfied with the answers.
>
> Before providing specific clarifications to your points, let us emphasize that our work should mostly be appreciated for the theoretical contributions. The experiments serve as a proof-of-concept to demonstrate that SPOIL can be applied in practice and is competitive with modern baselines such as IQ-Learn [1] (at least in the environments we tested, which are the same as the ones used for the offline experiments in [1]). Furthermore, our approach comes with strong performance guarantees that are rigorously proved under crisply stated conditions that are common in today’s RL theory literature.
>
> **Answers to the points raised in "Weaknesses"**
>
> **1. Explanations of the baselines**. Due to space limitations, we omit the descriptions of the baselines (IQ-Learn and PPIL) and refer to the respective original papers [1, 2] for the complete descriptions.
>
> **2. Regarding the scale of the experiments**. Given our compute constraints, these are the environments we could test on. The selected continuous control environments (CartPole, Acrobot, LunarLander) are standard benchmarks used for offline IL algorithm evaluation in the papers we compare against, including IQ-Learn and PPIL. We believe we did not make any unreasonably broad claims about state-of-the-art performance based on the experiments, but if some part of the text gave this impression, we are happy to revise it to ensure we do not make overstatements. We nevertheless would like to reiterate that we regard our theoretical results as the main contribution of this work, and those are state-of-the-art.
>
> **3. Regarding covariate shift**. We think that by covariance shift the reviewer means the fact that BC, when analyzed via a reduction to supervised learning like in [3, Theorem 2.1], has suboptimality which is $H^2 \epsilon_{SL}$ where $\epsilon_{SL}$ is the generalization error in supervised learning.
> While the idea of reducing a learning problem to a computational oracle (i.e. empirical risk minimization) is attractive, it has its limitations. Such reductions typically yield upper bounds on the sample complexity by translating an existing sample complexity bound for a simple setting (e.g., regression) to a more complicated setting (e.g., imitation learning) at the cost of some multiplicative factors (such as the $H^2$ in [3, Theorem 2.1]). This "black-box" approach does not always lead to tight guarantees, as one can plausibly do better by directly addressing the complicated setting (e.g., imitation learning) more directly, without pessimistically translating the worst-case errors from one setting to the next.
> Our work analyses bypasses the reduction to supervised learning and therefore the  covariate shift phenomenon. A second advantage of our analysis is that it does not require to assume that $\epsilon_{SL}$ is small to have small suboptimality with respect to the expert policy. It directly quantifies the number of expert demonstrations needed to achieve the desired suboptimality level $\epsilon$.
>
> **4. Regarding the intuition**. Thanks for the suggestion! We will add some additional intuitive explanations.
>
> **Answers to the questions**
>
> **1. Regarding the experimental results**. The first experiments show that BC and SPOIL perform similarly if the complexity of the expert policy class and state-action values class is comparable. On the other hand, SPOIL performs much better when the state-action value class is simpler than the expert policy class. The second set of experiments in Section 5.1 shows that SPOIL can be used in continuous state environments with neural network function approximation, achieving comparable or better performance than the baselines.
>
> **2. Differences with baselines**. PPIL and IQ-Learn differ considerably from our method because they use different objectives to update the Q-value (or a different "critic" update in deep RL jargon).
>
> **3. Significance of the improvements**. As stated, our algorithm has strong improvements when the Q-function class is much simpler than the expert policy class (see Experiment 1). In the control experiments, SPOIL is either better (see Acrobot) or on par with IQ Learn and PPIL.
>
> **4. Scale of the experiments**. We were limited to small-scale experiments due to the lack of computational resources, but we believe that these experiments were sufficient to convey the message of the paper. We do think that it would be interesting to conduct large-scale experiments with SPOIL for Atari, MuJoCo, and LLMs SFT, but we had to leave this for future work.
>
> We hope this response clarifies the contributions and context of our work. We believe our paper offers a relevant theoretical advance, supported by targeted and sound experiments. Thank you again for your time and constructive feedback.
>
> **References**
>
> [1] Garg, D., Chakraborty, S., Cundy, C., Song, J., and Ermon, S. (2021). Iq-learn: Inverse soft-q learning for imitation. Advances in Neural Information Processing Systems, 34, 4028-4039.
>
> [2] Viano, L., Kamoutsi, A., Neu, G., Krawczuk, I., and Cevher, V. (2022). Proximal point imitation learning. Advances in Neural Information Processing Systems, 35, 24309-24326.
>
> [3] Ross, S., and Bagnell, D. (2010, March). Efficient reductions for imitation learning. In Proceedings of the thirteenth international conference on artificial intelligence and statistics (pp. 661-668). JMLR Workshop and Conference Proceedings.

---

> > ### Comment · Reviewer_Mw96 · 2025-08-04
> >
> > Τhanks for your clarifications. Regarding some points you made in your response to the weaknesses identified:
> >  point (2) These evaluation environments were in fact used in such state-of-the-art works, however they were used as a proof of concept and intuition (see e.g. IQ-Learn) to then also provide more concrete evaluations using challenging MuJoCo and Atari games.
> > point (3) Interesting discussion, should be included in the paper.
> >
> > In general, the points you are making should have appeared in the paper in the first place.
> > The fact that experiments are in toy domains only, remains a major weakness.
> > I do understand that computational limitations might have been an issue.
> > For all these reasons, and following your clarifications , I am willing to raise my mark to "4", but not higher.

---

> > > ### Author Response · Authors · 2025-08-04
> > >
> > > Dear Reviewer,
> > >
> > > Thanks for engaging in the discussion and for the positive evaluation. We will add the content of our discussion in the final version.
> > >
> > > Best, Authors

---

### Official Review · Reviewer_UYBH · 2025-07-02

**Clarity:** 3
**Significance:** 2
**Originality:** 2
**Rating:** 4
**Confidence:** 2

**Summary:**

This paper studies the problem of offline imitation learning in MDP given a dataset of state action pairs of expert policy. This paper relaxes the assumption of expert policy belonging to a known class to the learner and introduces a new algorithm for the class of linear $Q^{\pi}$-realizable MDPs that guarantees to match expert performance up to $\epsilon$ with access to $\mathcal{O}(\epsilon^{-2})$. Furthermore, they extend their results to non linear setting but with a worse sample complexity. Finally, they conduct empirical evaluations on their algorithms.

**Questions:**

Questions:
- why on Figure 1, the BC agents doesn't improve much with complex policy class?
- can you run IQ-Learn in linear Q environments as well to see its performance against proposed algorithms?

**Ethical Concerns:**

["NO or VERY MINOR ethics concerns only"]

**Final Justification:**

This paper is both theoretical and empirical solid.
My questions regarding comparing against IQ Learn in linear setting is resolved, and the proposed algorithms demonstrate superior performance. Hence I recommend acceptance.

**Limitations:**

yes

**Quality:**

3

**Strengths And Weaknesses:**

Strengths:
- Quality: this paper is technically sound with support of theory and experiments.
- Clarity: paper is well presented with assumptions, theorems, algorithms and experiments.
- Significance: This paper relaxes a common assumption that expert policy is known to the learner and provides algorithms in both linear and non lineaner $Q^{\pi}$-realizable MDPs with theoretical guarantees.

Weaknesses:
- Originality: the novely mainly lies in the proof, the design of the algorithm is primal-dual style, which is similar to previous work of "Iq-learn: Inverse soft-q learning for imitation"

---

> ### Author Rebuttal · Authors · 2025-07-30
>
> Thank you for reviewing our work and for your positive evaluation. We are happy to address your questions below.
>
> **Regarding the performance of BC**. While the BC agent successfully minimizes the training loss, the dataset is not large enough for it to generalize well. Without discovering the simpler underlying linear structure of the environment, it cannot select good actions in states that were not (or sparsely) represented in the expert data, hence its poor performance.
>
> **Regarding a comparison with IQ-Learn**. In response to your question, we have run some additional experiments and implemented IQ Learn for both settings corresponding to two expert policies (the softmax linear policy and the neural network policy). We used a linear Q-function class for both SPOIL and IQ-Learn, as the environment has this structure. The results below show that, like SPOIL, IQ-Learn's performance is unaffected by the complexity of the expert class. However, SPOIL still reaches a higher cumulative reward in this environment.
>
> To ensure this is a fair comparison, we used the same learning rate for the policy updates in SPOIL and IQ-Learn. Moreover, we tested different choices for IQ-Learn’s critic learning rate, and we report here the best results we could obtain. The results are in the following tables, where we report the expected cumulative reward achieved by the three algorithms after the number of epochs reported in the column headers.
>
> |        $\pi_E \in \Pi^E_{\mathrm{lin}}$          | Epoch 10 | Epoch 50 | Epoch 100  | Epoch 250
> | :------------ | :-------------- | ---------------: | ---------------: |  ---------------: |
> | SPOIL   |   53.78 ± 0.15    |    61.68  ± 0.90    | 65.37 ± 2.09  | 67.62 ± 2.86   |
> | BC   |   52.78 ± 0.20    |    56.26  ± 0.89    | 59.37 ± 1.52  | 63.98 ± 2.40   |
> | IQ Learn   |   52.61 ± 0.26    |    56.20  ± 1.85    | 59.08 ± 2.96  | 61.57 ± 3.62   |
>
> |        $\pi_E \in \Pi^E_{\mathrm{NN}}$          | Epoch 10 | Epoch 50 | Epoch 100  | Epoch 250
> | :------------ | :-------------- | ---------------: | ---------------: |  ---------------: |
> | SPOIL   |   53.64 ± 0.20    |    60.35  ± 1.73    | 63.06 ± 2.99  | 64.71 ± 3.77   |
> | BC   |   51.74 ± 0.09    |    52.30  ± 0.23    | 52.72 ± 0.40  | 52.99 ± 0.52   |
> | IQ Learn   |   52.56 ± 0.26    |    55.66  ± 1.58    | 58.10 ± 2.61  | 60.29 ± 3.32   |
>
> Thank you again for your feedback. We are happy to discuss further if needed.

---

> > ### Comment · Reviewer_UYBH · 2025-08-05
> >
> > Thanks for the extra careful designed experiments that compares SPOIL and IQ-Learn to address my questions. Due to demonstrated empirical performance of the proposed algorithm, I decide to remain positive towards this submission.

---

> > > ### Author Response · Authors · 2025-08-05
> > >
> > > Dear Reviewer,
> > >
> > > Thank you for your answer and for confirming your positive evaluation.
> > >
> > > Best, Authors

---

### Official Review · Reviewer_dNPr · 2025-07-03

**Clarity:** 3
**Significance:** 3
**Originality:** 3
**Rating:** 5
**Confidence:** 3

**Summary:**

This paper considers the problem of imitation learning when the expert policy may not be realizable by the policy class at hand. The authors propose an alternative view via Q-functions, where instead the core structural assumption made is that any policy's corresponding Q function is realizable. Under this structural hypothesis, imitation learning can be cast as a Q-learning-like problem without directly imposing policy realizability. In particular, successfully solving the saddle-point problem of minimizing a policy that achieves low worst-case Q-error with respect to the expert. Sample complexity bounds are provided for this alternate procedure and simple numerical experiments demonstrate it can outperform naive behavior cloning.

**Questions:**

It would be helpful to clarify some of the above points under Weaknesses. Besides that, I have a question out of curiosity: when lacking reward labels, is this method compatible with inverse reinforcement learning? If not, what are the potential theoretical barriers?

**Ethical Concerns:**

["NO or VERY MINOR ethics concerns only"]

**Final Justification:**

This paper introduces some interesting theory-oriented ideas for understanding imitation learning via induced Q-functions rather than searching directly over policies. This has certain implications when the Q-class is "simpler" than the policy class.

There are some concerns expressed about the scale of the experiments, but I see the experiments mostly serving the purpose of justifying the theoretical mechanisms presented in the paper more so than promote a ready-for-deployment pipeline, and therefore am satisfied with them (including the ones presented in the rebuttal).

I think the (RL) theory community may find use of these results, and therefore recommend accept (5).

**Limitations:**

Yes

**Paper Formatting Concerns:**

None.

**Quality:**

3

**Strengths And Weaknesses:**

Overall, this paper introduces an interesting alternative perspective to imitation learning, where direct realizability of the expert policy can be avoided by instead considering the induced Q-functions of the policies. The analysis and interpretations in this paper make me lean toward acceptance.

### Strengths

The paper tackles a tenuous aspect of imitation learning theory in policy realizability. It is demonstrated that Q-realizability can be somewhat more natural of an assumption. Given the imitation learning objective is to achieve low reward suboptimality, the performance difference lemma demonstrates it suffices to mimic the Q function of the expert. The resulting objective function and ensuing analysis are to my knowledge novel and somewhat appealingly straightforward, which indicates that the ideas in this paper may be useful downstream.

### Weaknesses

The overall observation that imitation learning can be reduced to a fitting Q-functions is not necessarily new, see e.g. [1] which is discussed in the appendix related work. However, considering the similarities in the key object of study, these works should be explicitly discussed in the main paper.

Though the proposed method presents various advantages to other offline imitation learning methods, it is mentioned that policy-based methods may see advantages in other settings where the policy class is simple and the environment is difficult. It may be pedagogical for the camera-ready to provide a simple example where this is the case. Orthogonal to this nuance, a rather trivial observation of when policy-based methods such as BC are advantageous is when expert demonstrations do not carry reward labels, which deserves a bit of discussion, as a potential advantage of IL over RL is that it can be agnostic to rewards (and as such reward optimality).

As a bibliographic note, the paper [2] mentioned in the appendix related work, which considers imitation learning over continuous action spaces, demonstrates exponential-in-horizon error cannot be improved in general, actually includes any algorithm including reward-based ones, as long as the demonstration distribution is unchanged (i.e. offline IL).

[1] Swamy et al. "Of Moments and Matching: A Game-Theoretic Framework for Closing the Imitation Gap"

[2] Simchowitz et al. "The Pitfalls of Imitation Learning when Actions are Continuous"

---

> ### Author Rebuttal · Authors · 2025-07-30
>
> Thank you for your positive review, encouraging words, and insightful feedback! We answer the points you raised below.
>
> **Regarding some related work**. We will move the detailed comparison with Swamy et al. [1] from the appendix to the main body in the next version (taking advantage of the additional page allowed after acceptance).
>
> Moreover, we can also clarify the comparison with Simchowitz et al. [2]. As you noted, their work shows an unavoidable exponential-in-horizon error for offline IL *in general*. However, as we write in lines 540-545, their lower bound can be broken if the environment has structure. Although our algorithm is presented for discrete actions, we believe it could be adapted to continuous action problems by replacing the Mirror Ascent step in the policy update with continuous exponential weights (or discretizing the action set to a fine level).
>
> **Regarding a scenario favorable to BC**. As you suggested, we have run an experiment in which using BC should be more sensible than using SPOIL. More specifically, we consider a linear MDP (a special case of linear $Q^\pi$-realizable MDPs) with features of dimension $d=3$. The expert class, denoted $\Pi^E_{\mathrm{lin, small}}$, is the class of softmax linear policies with features corresponding to the linear MDP features, meaning BC is given the most compact representation possible. On the other hand, the Q-function class, denoted $\mathcal{Q}_{\mathrm{lin, large}}$, is made artificially complex: we take the linear MDP features and add a set of $20 d$ redundant features. It follows that the complexity of the Q-function class used for SPOIL is larger than the complexity of the policy class used for BC, and therefore BC is expected to outperform SPOIL on this instance.
>
> The following table reports the expected cumulative reward for the policies learned by SPOIL and BC after the number of epochs given in the column header. Notice that BC achieves a larger expected cumulative reward, as expected in this case.
>
> |        $\pi_E \in \Pi^E_{\mathrm{lin,small}}, Q^\pi \in \mathcal{Q}_{\mathrm{lin,large}}$          | Epoch 10 | Epoch 50 | Epoch 100  | Epoch 250
> | :------------ | :-------------- | ---------------: | ---------------: |  ---------------: |
> | SPOIL   |   64.67 ± 0.16    |    70.95  ± 2.43    | 81.51 ± 6.50  | 90.36 ± 9.62   |
> | BC   |   70.66 ± 2.06    |    83.37  ± 5.90    | 88.51 ± 7.32  | 92.53 ± 8.57   |
>
> **Comparison with inverse reinforcement learning:** Inverse Q-learning methods, such as SPOIL, have distinct trade-offs with inverse reinforcement learning (IRL).
> - On the "pro" side, SPOIL operates under weaker structural assumptions compared to IRL methods (e.g., $Q^\pi$-realizability instead of the more restrictive linear MDP assumption often used in IRL), and works entirely offline, whereas IRL methods typically require online environment interaction.
> - On the other hand, SPOIL requires state-action pair observations. In scenarios with more limited information, such as observing only reward features, some IRL methods (e.g., [3, 4]) are more applicable. Both SPOIL and BC are reward-agnostic in that they learn from demonstrations without needing reward labels. The key difference is that SPOIL adopts a value-based approach to match performance, while BC uses supervised learning to match behavior.
>
> Thanks again for your feedback. We are available for any further questions.
>
> **References**
>
> [3] Ziebart, B. D. (2010). Modeling purposeful adaptive behavior with the principle of maximum causal entropy. Carnegie Mellon University. Section 6.2.
>
> [4] Syed, U., and Schapire, R. E. (2007). A game-theoretic approach to apprenticeship learning. Advances in neural information processing systems, 20.

---

> > ### Comment · Reviewer_dNPr · 2025-08-04
> >
> > Thank you for the clarifications. The additional experiment and discussion about IRL are helpful. I maintain my positive evaluation.
> >
> > Since I work with continuous control, I will note a few cautions that may aid the authors downstream if they decide to tackle that setting.
> >
> > 1. Typical RL-theoretic structural assumptions (e.g. linear/low-rank MDP) don't necessarily make a lot of sense for dynamical systems, which typically start with deterministic policies + dynamics, then perhaps zero-mean stochasticity on top of that (well-motivated by physical systems).
> >
> > 2. Discretizing action spaces can't subvert general lower bounds, e.g. in the aforementioned "Pitfalls..." paper, since without dynamics/transition kernel knowledge, the discretization scheme requires in general exp(dim(actions)) bins to distinguish between stable and unstable policies.
> >
> > 3. Value-based IL methods are interesting, and typical RL assumptions (e.g. Lipschitz Q-class) do enable nice guarantees, such as in Swamy et al. However, the subtlety in continuous control is that Q-fxn candidates implicitly index $(\pi, f)$ a policy-dynamics pair (since in offline we cannot know the true f to roll out). As also pointed out in "Pitfalls...", subexp(horizon)-Lipschitz Q-fxns correspond to stable $(\pi, f)$ pairs --- however, stable $(\pi, f)$ need not imply $(\pi, f^\star)$ is stable, i.e. the "true" Q-function corresponding to rolling out $\pi$ on the true dynamics need not be Lipschitz. Therefore, restricting to nice Q-classes does not immediately eliminate "bad" policies.
> >
> > Hope this might be helpful if the authors decide to consider expanding the theory scope to continuous control settings.

---

> ### Author Response · Authors · 2025-08-05
>
> Dear Reviewer,
>
> Thanks a lot for your answer and for your support.
>
> The continuous action extension is indeed very interesting, we will think about it.
> Thank you for your helpful comments about this setting.
>
> Best,
> Authors

---

### Decision · Program_Chairs · 2025-09-17

**Decision:**

Accept (poster)

**Comment:**

The paper introduces a new bound for the performance of a policy learned in the offline imitation learning setting. It makes an important structural assumption (for a policy, the q function of a state-action tuple is a dot product of some feature of state-action tuple and some vector). It also provides a (weaker) result under less restrictive assumptions.

The main strength of the paper is it addresses an important subproblem in the offline IL community.

The main weaknesses is the lack of extreme novelty (novelty is sufficient overall though).

Overall, this paper should be accepted because it solves an important problem well.

The discussion focussed on: (1) novelty (2) the fact that we don't observe rewards (3) covariate shift.